# IM30 IDPs form a membrane-protective carpet upon super-complex disassembly

Benedikt Junglas[1], Roberto Orru[2], Amelie Axt[3,4], Carmen Siebenaller[1], Wieland Steinchen[5], Jennifer Heidrich[1], Ute A. Hellmich[1,6], Nadja Hellmann[1], Eva Wolf[2,7], Stefan A. L. Weber[3,4] & Dirk Schneider[1✉]

Members of the phage shock protein A (PspA) family, including the inner membrane-associated protein of 30 kDa (IM30), are suggested to stabilize stressed cellular membranes. Furthermore, IM30 is essential in thylakoid membrane-containing chloroplasts and cyanobacteria, where it is involved in membrane biogenesis and/or remodeling. While it is well known that PspA and IM30 bind to membranes, the mechanism of membrane stabilization is still enigmatic. Here we report that ring-shaped IM30 super-complexes disassemble on membranes, resulting in formation of a membrane-protecting protein carpet. Upon ring dissociation, the C-terminal domain of IM30 unfolds, and the protomers self-assemble on membranes. IM30 assemblies at membranes have been observed before in vivo and were associated with stress response in cyanobacteria and chloroplasts. These assemblies likely correspond to the here identified carpet structures. Our study defines the thus far enigmatic structural basis for the physiological function of IM30 and related proteins, including PspA, and highlights a hitherto unrecognized concept of membrane stabilization by intrinsically disordered proteins.

[1] Department of Chemistry, Biochemistry, Johannes Gutenberg University Mainz, 55128 Mainz, Germany. [2] Institute of Molecular Physiology, Johannes Gutenberg University Mainz, 55128 Mainz, Germany. [3] Max Planck-Institute for Polymer Research, 55128 Mainz, Germany. [4] Institute of Physics, Johannes Gutenberg University Mainz, 55099 Mainz, Germany. [5] Philipps-University Marburg, Center for Synthetic Microbiology (SYNMIKRO) and Department of Chemistry, 35032 Marburg, Germany. [6] Centre for Biomolecular Magnetic Resonance (BMRZ), Goethe-University Frankfurt, 60438 Frankfurt, Germany. [7] Institute of Molecular Biology (IMB), 55128 Mainz, Germany. ✉email: Dirk.Schneider@uni-mainz.de

The inner membrane-associated protein of 30 kDa (IM30), also known as vesicle inducing protein in plastids (VIPP1), is a protein conserved in chloroplasts and cyanobacteria, where it is involved in thylakoid membrane (TM) biogenesis and/ or maintenance[1–15]. A striking feature of IM30 protein family members is the ability to form large homo-oligomeric super-complexes with masses exceeding 1 MDa[16–18]. With transmission electron microscopy (TEM), a ring-like organization with a distinct spike architecture has been observed for these complexes[19]. Besides ring structures, rod-like particles have also been observed that might form via ring stacking[17,19–24]. Although no high-resolution structure of IM30 is currently available, the IM30 structure appears to resemble the structure of its supposed ancestor, the bacterial phage shock protein A (PspA)[1,16,20]. For both protein monomers, six α-helical segments have been predicted. In addition, IM30 contains an extra helix at its C-terminus. A coiled-coil hairpin structure of two extended helices (helices 2 and 3) likely represents the structural core of both, PspA and IM30[25]. The structure of this fragment has recently been solved and was used as a template for the prediction of the IM30 full-length tertiary structure[19]. IM30 binds to membranes, where it forms assemblies, as it has been observed in in vivo studies with GFP-labeled IM30. In cyanobacteria and chloroplasts, such assemblies form dynamically at TM margins[3,9,11]. It has been proposed that these assemblies are involved in membrane protection/stabilization[4,26], due to the membrane protective effects of IM30 observed in *Arabidopsis thaliana* chloroplasts[9,10,13,14]. Importantly, the IM30 rings can adsorb to negatively charged membranes[12], albeit the formation of large ring structures is clearly not crucial for membrane binding in vitro, as small IM30 oligomers bind to negatively charged membranes with even higher affinity than IM30 rings[27]. Therefore, the interaction of IM30 with membranes may thermodynamically drive ring disassembly on membrane surfaces and thus entail disassembly of IM30 rings on the membrane surface. However, the question whether and how IM30 rings may disassemble during membrane interaction is completely unsolved so far.

Here we show that IM30 rings disassemble on membrane surfaces upon binding, and disassembly of IM30 rings involves unfolding of the predicted helices 3–7 located in the C-terminal half of the protein. Intrinsically disordered IM30 can bind with high affinity to membrane surfaces where protomers assemble to form a surface-covering carpet structure that stabilizes membranes.

## Results

### IM30 super-complexes disassemble upon membrane binding and rearrange into carpet-like structures. Supporting the hypothesis that IM30 rings undergo a structural rearrangement upon membrane binding, we observed differences in the trypsin-digestion pattern of IM30 in absence vs. presence of phosphatidylglycerol (PG)-containing liposomes (Supplementary Fig. 1). Yet, these observations do not allow to clearly discriminate between rearrangements of the IM30 structure, shielding of IM30 regions due to membrane binding, or a combination of both. To probe potential ring disassembly upon membrane binding more directly, we next employed the FRET signal established between CFP and Venus-labeled IM30 monomers incorporated in IM30 rings. While we observed decreasing FRET in the presence of PG liposomes (Fig. 1a), indicating a change in the relative distance between individual monomers upon membrane binding, these FRET changes remained minor and leveled off at high lipid concentrations. Thus, some structural changes potentially occur,

possibly limited disassembly; yet, on average the monomers appear to stay in close contact on the membrane surface.

To visualize IM30 bound to PG supported lipid bilayers (SLBs), we next used atomic force microscopy (AFM). While the expected ring structures were apparent when IM30 WT was bound to a mica surface (Supplementary Fig. 3a, b), flat carpet-like structures became visible on the membrane surfaces after incubating a PG bilayer with IM30 WT (Fig. 1a). These structures cover an area of several hundred nm², have a rough and uneven surface, and a height of 0.7–1.9 nm (average height: ~0.9 nm). As IM30 rings have a height of 13–15 nm[19], these carpets do clearly not form simply via lateral association of multiple IM30 rings on a membrane surface, again suggesting the disassembly of membrane-bound IM30 rings into smaller oligomers and their rearrangement on the membrane surface. To investigate whether the formation of the observed carpet structures requires the preceding formation of IM30 rings, we made use of an oligomerization-impaired mutant (IM30*). At suitable NaCl concentrations, IM30* exclusively forms dimers (Supplementary Fig. 2). Since the IM30* carpets are alike those formed by the WT protein, we conclude that carpet formation by IM30 does not per se require ring formation (and subsequent dissociation) (Fig. 1b). Noteworthy, carpet formation was not observed when IM30 WT or IM30* were incubated on mica surfaces, i.e., in absence of a membrane (Supplementary Fig. 4). As cyanobacterial and chloroplast membranes typically contain about 40% negatively charged membrane lipids[15], we additionally analyzed the formation of carpet structures on PC: PG (60:40) membrane surfaces (Supplementary Fig. 5). Yet, IM30 WT, as well as IM30*, disassemble and form carpet structures also on this membrane surface, excluding the possibility that the observed carpet formation was induced by the highly charged membrane surface.

### IM30 carpets protect destabilized liposomal membranes. Due to the importance of IM30 for TM maintenance, we wondered whether formation of the carpet structures might have functional consequences, e.g., for the membrane integrity. We, therefore, compared the ability of IM30 WT super-oligomeric rings vs. IM30* dimers to maintain a proton gradient across a membrane, using a fluorescence-based proton leakage assay. Here, proton flux into the liposomes was measured as a decrease in ACMA fluorescence[28,29]. Only a small proton flux was measured with untreated PG liposomes (control, Fig. 1d), whereas the addition of 6% (v/v) isopropanol weakened the membrane integrity considerably and increased the proton permeability of the liposomal membranes (negative control, Fig. 1d). Addition of IM30 WT and IM30* led to a reduced proton permeability of the liposomes, with IM30* showing enhanced reduction, possibly because the energetic cost of super-complex disassembly did not have to be paid. When we compared membrane binding of IM30* with IM30 WT rings over 20 min, the binding kinetics between the two proteins differed (Fig. 1e). Binding of the dimeric IM30* reached equilibrium earlier than the WT protein. This indicates that membrane binding of IM30 WT rings is slower than binding of smaller IM30* oligomers. The faster binding of IM30* could just be due to a larger number of particles adsorbing to the membranes compared to the rings, at identical monomer concentration. Only upon ring disassembly, full membrane adsorption of IM30 WT monomers is accomplished. Taken together, the interaction of IM30 with negatively charged membranes involves an initial membrane-binding step (potentially involving minor structural changes and ring destabilization), subsequent ring disassembly and rearrangement to carpet structures that form a protective layer on the membrane.

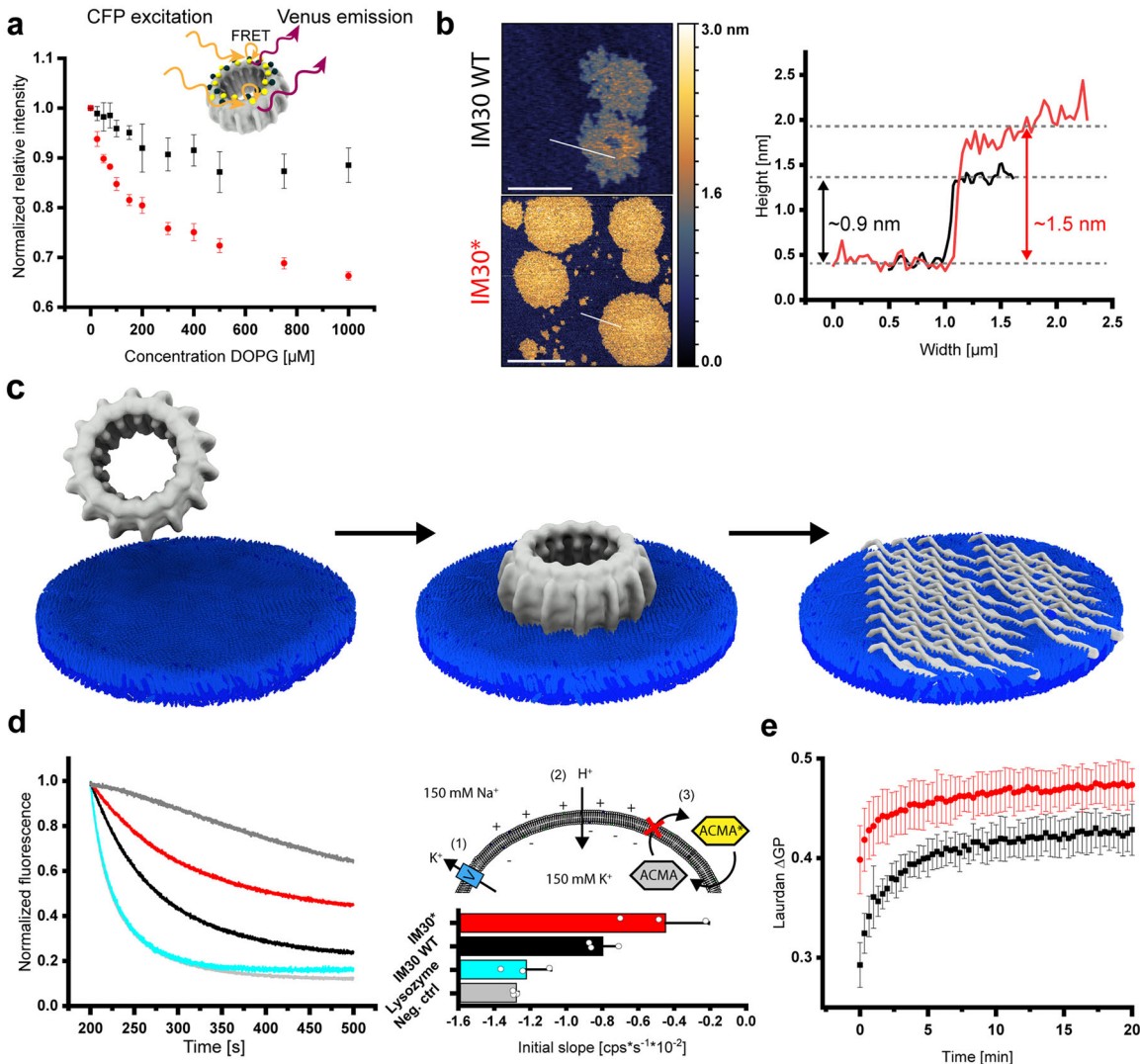

**Fig. 1 Binding of IM30 to negatively charged membrane surfaces results in ring disassembly, carpet formation, and membrane stabilization. a** FRET was measured using IM30 rings containing both, IM30-CFP and IM30-Venus monomers (red). The normalized relative fluorescence intensity (see "Methods") is plotted against the DOPG concentration. The intensity decreases with increasing DOPG concentrations, indicating an increasing average distance between the monomers upon interaction with membranes. Noteworthy, the fluorescence characteristics of the fluorophores alone change only slightly upon membrane binding, resulting in an apparently altered FRET (black). The error bars represent SD, $n = 5$. **b** The structure of IM30 WT and IM30* bound on a PG bilayer was imaged via AFM (the false-color ruler indicates the heights in the images). Both IM30 variants form carpet-like structures. The height-profiles (white section lines in the images) of the carpet-like structures indicate similar heights of IM30 WT (black line) and IM30* (red line) carpets. Determined heights are in the range of 0.7–1.9 nm). Single coherent IM30* carpets have increased dimensions, which leads to edges appearing rounder than the fractal-like shape of IM30 WT carpets. Scale bar: 1 μm (upper panel) and 3 μm (lower panel). **c** IM30 appears to initially bind to the membrane as a ring, followed by disassembly into small oligomers/monomers and rearrangement to a carpet-like structure. The ring structure was taken from EMD:3740[19]. **d** ACMA fluorescence was used to monitor proton flux across DOPG membranes. Untreated liposomes were slightly permeable for protons (positive control, dark gray), whereas DOPG liposomes have high proton permeability in presence of 6% isopropanol (negative control, light gray). Lysozyme, which was used as a control (cyan), had no effect on the proton permeability. In presence of IM30 WT (black), the proton permeability of isopropanol-treated DOPG liposomes was reduced. This effect was much stronger in presence of IM30* (red). For quantitative analysis, the initial slope of the fluorescence changes was evaluated. Error bars represent SD ($n = 3$). **e** Lipid-binding of IM30 WT (black) and IM30* (red) to PG liposomes was determined via monitoring Laurdan fluorescence changes. IM30* affects the Laurdan fluorescence emission characteristics (ΔGP) much faster than the WT protein. Error bars represent SD ($n = 3$).

**IM30 is highly flexible when not organized in super-complexes.** As the dimeric IM30* protein appears to be hyper-functional in the proton leakage assay (Fig. 1d), we next elucidated the structure and shape of small IM30 oligomers using SAXS (small-angle X-ray scattering) coupled to size exclusion chromatography (SEC-SAXS). The SEC elution profile and the averaged scattering intensity confirmed a high sample homogeneity (Fig. 2a, b and Supplementary Fig. 10a). Analysis of the SAXS data resulted in a

molecular mass of 63.2 ± 5.2 kDa, as expected for an IM30* dimer (Supplementary Fig. 10a). We obtained a radius of gyration of 6.13 ± 0.05 nm and the pair distance distribution yielded a $D_{max}$ of 26 nm (Fig. 2c and Supplementary Fig. 10b), indicating that IM30* adopts an elongated shape. When we compared our SAXS data with the structures of other proteins in a dimensionless Kratky-plot, it became apparent that IM30* does not have a well-defined, compact and spherical shape, but an extended and

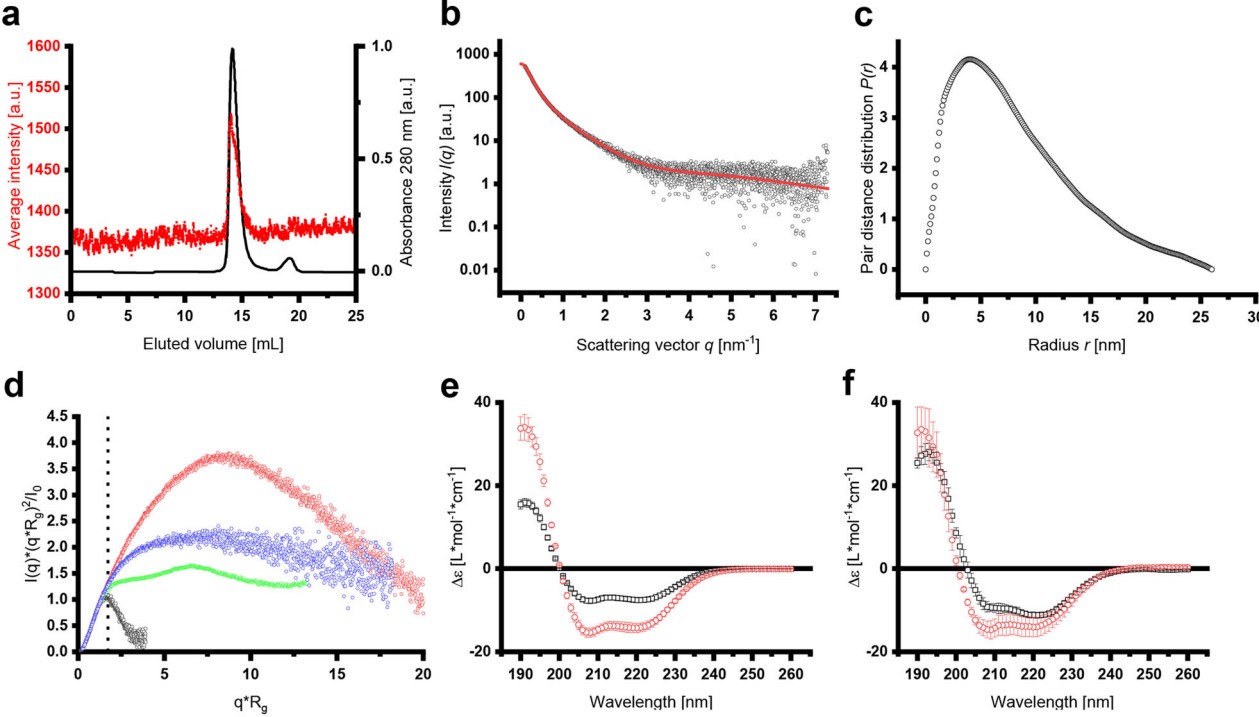

**Fig. 2 SEC-SAXS analyses of IM30* dimers. a** The average SAXS intensity (red dots) is compared to the normalized absorbance at 280 nm (black line) over the whole elution volume. **b** The scattering intensity after buffer subtraction was plotted against the scattering angle $q$. The red line represents the fit of the data for the pair distance distribution analysis by GNOM ($\chi^2 = 1.0392$). **c** The pair distance distribution analysis in the range of $q = 0.0929–7.2965$ $nm^{-1}$ and forcing to 0 at $D_{max} = 26$ nm gave $I_0 = 601.3 \pm 4.5$ $cm^{-1}$ and $R_g = 6.86 \pm 0.07$ nm (total quality estimate from GNOM 0.59). **d** A dimensionless Kratky-plot was used to compare the scattering data obtained with IM30* and other proteins. Apparently, the Kratky curve of IM30 dimers lies in between the curves of the unfolded lysine riboswitch protein and the Plakin domain of Human plectin, which has an extended protein shape, clearly implying an extended and somewhat flexible structure of IM30* dimers. The dashed line indicates $q * R_g = \sqrt{3}$. Black dots: Lysozyme (SASDA96)[72]. Red dots: Plakin domain of human plectin (SASDBC4)[73]. Green dots: Unfolded lysine riboswitch (BIOISIS ID:2LYSRR)[74]. Blue dots: IM30*. **e** The CD spectrum of IM30* (black squares) showed the typical characteristics of a mainly α-helical protein, i.e., pronounced minima at 222 and 208 nm. Yet, the amplitudes of the minima at 222 nm and 208 nearly doubled upon addition of 8 M TFE (red circles), which is known to induce α-helical structures in proteins/peptides. This implies that IM30* is highly unstructured. Error bars represent SD ($n = 3$). **f** The amplitudes of the minima at 222 nm and 208 nm of IM30 WT (black squares) only slightly increase upon addition of TFE (red circles), confirming the expected high content of α-helical structures. Error bars represent SD ($n = 3$). Based on the CD-spectra, the α-helix content of IM30* (**e**) was calculated to be ~57%, which is considerably lower than the reported and predicted α-helix content of IM30 WT of ~80%[16–18]. In presence of TFE, both proteins reach about 100% α-helix content.

somewhat flexible structure with a high content of unstructured regions (Fig. 2d). Indeed, CD analyses showed that ~40% of IM30* is unstructured. In contrast, the IM30 WT protein has an α-helix content of ~80% (Fig. 2e, f), in line with the IM30 structural model proposed by Saur et al.[19].

**IM30 dimers have a disordered N-terminus and C-terminal domain**. To assess the inherent structural flexibility of IM30* dimers in greater detail, we carried out limited proteolysis and observed a single stable IM30 fragment of ~17 kDa, which contained parts of helix 1 to approximately half of helix 4 (Fig. 3a). In conclusion, helix 1 and helix 4–7 appear to be flexible in the IM30 monomer, whereas helices 2 and 3 form a stable structure. To more clearly define the disordered regions, we next used hydrogen-deuterium exchange (HDX) measurements coupled to LC-MS on the IM30* and IM30 WT proteins. The results were mapped on the structural model of the monomer suggested by Saur et al.[19] (Fig. 3b). The HDX data confirmed that helices 2 and 3a in the suspected stable core region indeed exhibited only weak H/D exchange in both IM30 WT and IM30*. As expected, the flexible linker between helix 6 and 7 showed high H/D exchange in both variants, as did helix 7 (Supplementary Fig. 8a, b). The major structural difference between IM30* and IM30 WT lies in the region of the predicted helices 1, 3b, 4, and 5/6, where the WT

protein showed less H/D exchange than the mutant (Fig. 3b and Supplementary Fig. 8). Likely, IM30* has an unstructured N-terminal domain (helix 1) and a mostly unstructured C-terminal domain (helices 3b-7), in excellent agreement with the limited proteolysis data (Fig. 3a). Indeed, using CD and 1D-$^1$H-NMR spectroscopy of the isolated IM30_H3b-7 fragment, we could confirm that this region is completely unstructured (Supplementary Fig. 9). Hence, IM30* dimers have an unstructured C-terminal domain, while IM30 is highly structured when organized in higher-ordered (ring) structures (Fig. 2e, f and Supplementary Fig. 8b). Thus, as IM30 WT forms nearly exclusively large super-complexes in solution[17,19], the formation of such higher-ordered structures appears to induce folding of otherwise intrinsically disordered IM30 regions.

To generate a structural model of IM30 monomers that includes the highly flexible nature of the IM30* C-terminus, we used a fragmentation-based modeling approach based on SAXS envelopes, starting from the structural model described in Saur et al.[19]. The SAXS envelopes were calculated as described in detail in Supplementary Fig. 10. We used the available X-ray structure of the PspA helix 2/3 fragment[25] as a rigid structural core and rendered the remaining parts of the structure as highly flexible and/or unstructured (as identified above). The resulting models and their respective SAXS envelopes are shown in Supplementary

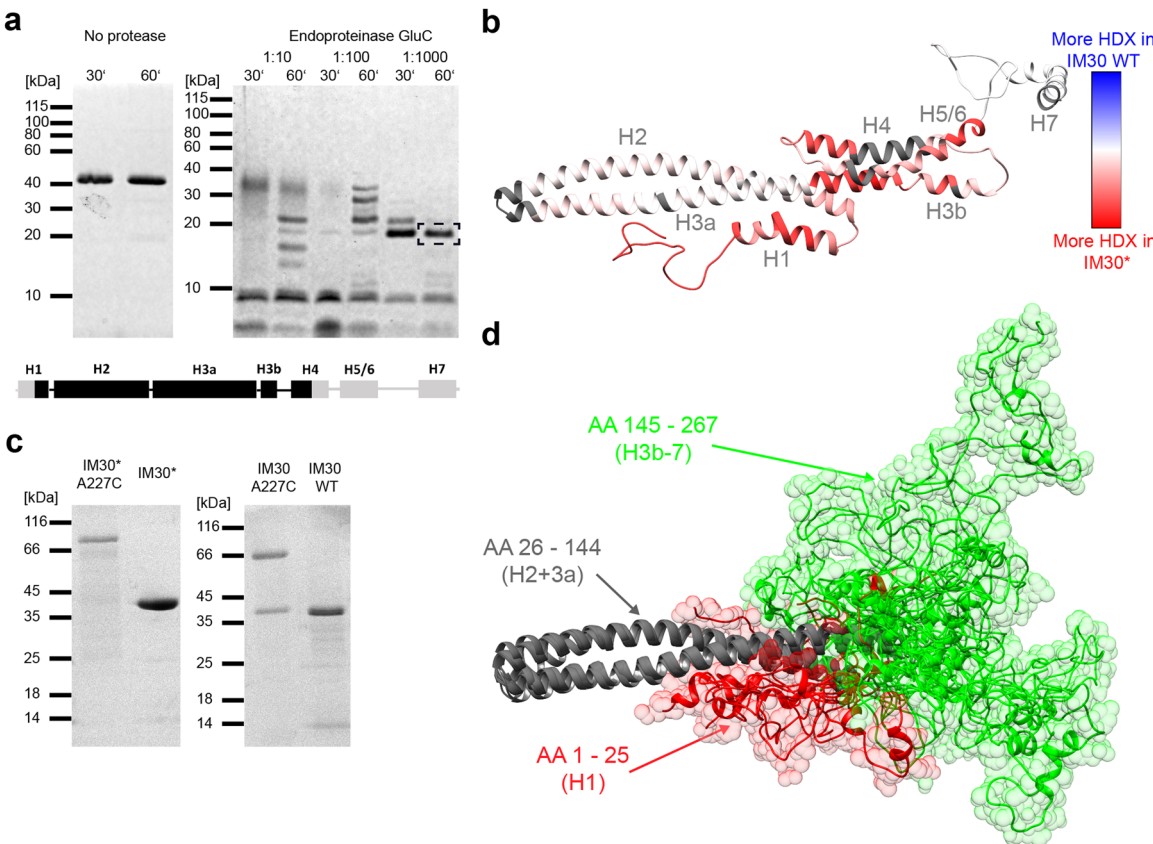

**Fig. 3 IM30\* dimerizes via its unstructured C-terminus.** Limited proteolysis of IM30\*: IM30\* was incubated with the endoproteinase GluC at different enzyme-to-substrate ratios for 30 or 60 min, respectively. The peptide patterns were analyzed via SDS-PAGE. The band highlighted with a black box was analyzed by mass spectrometry. Below, the predicted secondary structure of IM30 is shown, whereby each box represents an α-helical segment. After treatment with endoproteinase GluC, a stable helix 2/3 fragment was identified (with a sequence coverage of ~60%). For more details, see Supplementary Fig. 7. **b** The difference of relative HDX between IM30\* and IM30 WT after 10 s mapped on the predicted IM30 monomer structure[19] revealed an increased flexibility of helix 1 and helices 3a-5/6 of IM30\* compared to the WT. A large part of helix 7 and the linker region between helix 7 and 6 is highly flexible in both variants. Dark gray regions mark sites where no peptides were detected in the HDX experiment, and thus no data is available. (For more details see Supplementary Fig. 8). **c** SDS-PAGE analysis of A227C mutants of IM30 WT and IM30\* in absence of reducing agents. **d** IM30 monomer models generated based on the SAXS data were aligned at helix 2 and 3a to visualize the flexibility of the C-terminal region. Helices 2 and 3a (amino acids 26–144) are depicted in gray, the C-terminal amino acids 145–267 (H3b-7) are colored in green, and amino acids 1–25 (H1) in red. The N-terminal region (red) fills a small volume, starting from the rim between helix 2 and 3 with an only small overlap between the N-terminal and the C-terminal regions. However, the C-terminal region (green) nearly fills the entire conformational space at the end of the structural core, with higher accumulation at the very end of helix 3a.

Fig. 12. All envelopes used are compatible with the experimental scattering pattern, fitting to a similar degree, and thus all calculated conformations likely represent actual IM30\* dimer structures. Each envelope can be considered as a snapshot of one possible conformer, indicating enormous flexibility in the dimer interface region. In Fig. 3d, the intrinsic flexibility is visualized by superimposing individual monomers from each SAXS-model, aligned at the structured core (helices 2 + 3) of the respective monomers.

To define the IM30 regions that mediate dimer formation, we next used SEC coupled multi-angle laser light scattering (SEC-MALS) and determined the oligomeric state of two truncated IM30 versions, representing the stable structural core (helices 2 + 3; IM30_H2-3a) and the intrinsically disordered C-terminus (helices 3b-7; IM30_H3b-7) (Supplementary Fig. 13). While IM30_H3b-7 likely formed dimers, IM30_H2-3a exclusively formed monomers under our experimental conditions. Thus, IM30\* dimerizes via residues located in the C-terminal domains, in line with our dimer models (Supplementary Fig. 12), whereas the N-terminal region could mediate membrane interaction. In fact, stable membrane interaction of the isolated helix 2/3

fragment has been demonstrated recently[30]. Based on our dimer models (Supplementary Fig. 12), the flexible linker between the regions assigned as helix 6 and 7 in the model predicted by Saur et al.[19] appears to be crucially involved in mediating contacts between two adjacent monomers. To validate the predicted role of this region in IM30\* dimerization, we created an IM30\* variant where Ala227, located in the extended linker between helix 6 and 7 (compare Supplementary Fig. 2a), was mutated to Cys. This mutant ran as a dimer on SDS gels after purification (Fig. 3c), which indicates that the regions between helix 6 and 7 of two adjacent monomers are indeed in close contact in IM30\* dimers. Noteworthy, also the IM30 WT protein formed covalently linked dimers, when we introduced the A227C mutation into the IM30 WT sequence (Fig. 3c). Thus, when not arranged in super-complexes, (at least) IM30 dimers have an intrinsically disordered C-terminal domain that clearly is involved in protein dimeriza-tion (as shown here). Furthermore, at reduced salt concentra-tions, this domain can also form higher-ordered oligomers (Supplementary Fig. 13c), and thus the isolated disordered region likely has multiple positions for binding other IM30 proteins, which facilitates self-assembly on membranes.

## Discussion

The core structure of IM30 proteins is the coiled-coil formed by the helices 2 and 3[15,19,30,31]. This structure is stable even in the isolated, monomeric helix 2-3a fragment[30]; thus, no additional interactions with other parts of the protein are required for secondary and tertiary structure formation of this fragment. In contrast, the isolated C-terminal region comprising helix 3b-7 is largely unstructured, albeit capable of forming stable dimers. If combined in the full-length WT protein, large oligomeric rings form, in which also the helix 3b-7 region appears to be mostly α-helical (Supplementary Fig. 8b). Thus, folding and homo-oligomerization of the h3b-7 region are interconnected, and inter-molecular interactions within the oligomer appear to induce the formation of α-helices. This interconnection is supported by the observation that the C-terminal region of the oligomerization-incompetent mutant IM30* remains unstructured even in the full-length protein, most likely because stabilizing interactions with neighboring protomers are largely reduced. Actually, the structure of the full-length IM30* protein resembles the sum of the two WT fragments. Thus, it is reasonable to assume that also in the full-length WT protein the C-terminal region is largely unstructured when the protein is not part of IM30 super-complexes.

While the isolated C-terminal region of IM30 oligomerizes, assembly of IM30 ring super-complexes involves additional interactions between other IM30 regions. In fact, we recently observed that the isolated helix 2/3 coiled-coil does have an intrinsic propensity to dimerize[30] but does not form higher-ordered super-complexes. Thus, interactions involving both, the helix 2–3 coiled-coil as well as (at least) the region containing helices 4–6, are required for ring formation[30].

Likely, the residues of the conserved FERM cluster located in helix 4 are crucially involved in mediating and/or stabilizing contacts between adjacent IM30 protomers in the ring. Weakening (or abolishing) these contacts via mutation of these residues has now enabled us to analyze the structure and activity of small IM30 oligomer, i.e., IM30* dimers. While the structure of the helix 2–3 core is mostly unaffected when the dimers are compared with the super-complexes (see Fig. 3b), the structure of the C-terminal helices 4–7 dramatically differs when IM30(*) monomers are not part of ring super-complexes. While the introduced Ala residues were expected to further promote α-helix formation[32], the C-terminal part of the here analyzed variant remains completely unstructured (Fig. 3).

Yet, the dimeric IM30* protein shows faster membrane binding and more efficient membrane protection than the WT protein (Fig. 1d, e). This observation is perfectly in line with the recent notion that the isolated helix 2–3 coiled-coil effectively binds to membrane surfaces[30]. As this coiled-coil is buried and involved in protomer–protomer interactions when IM30 monomers are part of higher-ordered ring structures[19,30], the WT protein can efficiently bind to membrane surfaces only upon ring disassembly. This crucial step in carpet formation is not required anymore in case of the dimeric IM30* protein, where the helix 2/3 coiled-coil is readily exposed to facilitate membrane binding. However, membrane binding of the helix 2–3 coiled-coil alone does not result in membrane protection, but rather in membrane destabilization[30]. This strongly suggests that the C-terminus is mainly responsible for the membrane-protecting activity of IM30, in line with the observation that the isolated C-terminus oligomerizes (Supplementary Fig. 13a). As shown here, IM30-mediated membrane protection is associated with the formation of carpet-like structures on the membrane surface (Fig. 1). These carpets form via association of IM30 protomers on the membrane surface, but not in solution, and involve interactions between the disordered C-terminal regions.

Disordered protein domains exhibit an increased surface area for interaction, which can be beneficial for interaction with multiple binding partners. Self-assembling IDPs (intrinsically disordered proteins) can form higher-ordered protein complexes, where disordered protomers undergo binding-induced folding during super-complex formation[33–35], which also appears to be the case when IM30 rings form in solution. Vice versa, IM30 rings appear to disassemble upon membrane binding and condensate into extended carpets on the membrane, which again requires interactions between the disordered C-termini. Noteworthy, while not observed here, carpet formation could also involve partial structuring of this region.

Protein self-assembly on membrane surfaces, resulting in membrane-covering protein structures, has been observed before, e.g., in case of Alzheimer's or Parkinson's disease[36,37]. Yet, here formation of protein assemblies on membrane surfaces results in membrane destabilization and rupture. In contrast, IM30 carpets suppress proton leakage in liposomes and thereby maintain the integrity of membranes, as previously suggested for its ancestor PspA, which is thought to form scaffold-like structures to cover large membrane areas and prevent leakage[38,39]. The idea of IM30 and PspA having similar membrane-stabilizing functions is in agreement with the observation that IM30 can functionally complement E. coli pspA null mutants[40]. This finding is also in line with the observation that IM30-overexpressing Arabidopsis thaliana strains display improved heat stress recovery[14] and that IM30 forms large assemblies at TMs in cyanobacteria under stress conditions[11,14]. These assemblies, which likely correspond to the IM30 carpet structures observed in the present study, have been identified in vivo to dynamically localize, preferably at stressed TM regions[3]. In fact, dynamic self-assembly is typically observed with IDPs, often involving liquid–liquid phase separation[33,41,42]. In contrast to the formation of membrane-less organelles in cells, induced by liquid–liquid phase separation of IDPs, demixing into a condensed and a protein-light fraction (i.e., carpets and unassociated but membrane-attached protomers) appears to take place on the membrane surface in case of IM30. Restricting protein-protein interaction to the membrane surface limits the degrees of freedom to a 2D surface, which likely increased the efficiency of carpet formation.

## Methods

**Expression and purification of IM30.** N-terminally His-tagged *Synechocystis* IM30 (IM30 WT; from *Synechocystis sp.* PCC 6803) was expressed in *E. coli* BL21 (DE3) using a pRSET6 based plasmid. Cells were resuspended in 50 mM NaPhosphate, 300 mM NaCl, 20 mM imidazole (pH 7.6) and lysed by sonification. IM30 was purified from the cleared lysate using Ni-Affinity chromatography[12]. IM30* (E83A, E84A, F168A, E169A, R170A, M171A), IM30_A227C (A272C) and IM30*_A227C (E83A, E84A, F168A, E169A, R170A, M171A, A227C) were generated via site-directed mutagenesis. The WT-fragments IM30_H2-3a (amino acids 22–145) and IM30_H3b-7 (amino acids 147–267) were generated by PCR cloning[30]. IM30-CFP and IM30-Venus were generated by restriction digestion and T4 ligation of the CFP/Venus fragments into pRSET *Syn*IM30 plasmids[43]. All IM30 variants were expressed and purified as described for the WT[12]. After isolation, the proteins were further purified by size exclusion chromatography (Superdex 200 16/60 HL, GE Healthcare) and eluted in 20 mM HEPES pH 7.6 at 8 °C. Peak fractions were pooled and concentrated by a centrifugal filter unit (PES membrane (PALL), 5000 g, 4 °C). Protein concentration was estimated by absorbance at 280 nm or 230 nm for the IM30_H3b-7, respectively.

**Size exclusion chromatography (SEC).** The oligomeric state of IM30* and IM30 fragments was analyzed using an ÄKTA basic system (GE Healthcare) with a Superose12 10/300 GL column (GE Healthcare) equilibrated with 20 mM HEPES pH 7.6 and 0, 50, 100, 150 or 300 mM NaCl at 8 °C. Protein elution was monitored at 280 nm. The column was calibrated using proteins of known molecular mass.

**SEC coupled multi-angle laser light scattering.** The oligomeric states of IM30_H2-3a and IM30_H3b-7 were analyzed by SEC coupled multi-angle laser light scattering (SEC-MALS). Protein solutions of IM30_H2-3a or IM30_H3b-7 in 25 mM HEPES, 125 mM NaCl, 5% glycerol (w/w) were analyzed at RT, using a

Superdex 200 Increase 10/300 GL column (GE Healthcare) equilibrated with 25 mM HEPES (pH 7.5), 125 mM NaCl, 5% glycerol (w/w) connected to an UV-Vis detector (BioRad UV 1806) and a MALS detector (Wyatt DAWN DSP) using an Agilent 1100 series pump. Protein elution was monitored by absorbance at 280 nm for IM30_H2-3a ($\varepsilon_{280} = 0.417$ cm$^2$ mg$^{-1}$) or 230 nm for IM30_H3b-7 ($\varepsilon_{230} = 2.747$ cm$^2$ mg$^{-1}$), respectively.

**Trypsin digestion of IM30**. 2.5 µM IM30 WT was incubated in absence or presence of 0.1 mM DOPG (dioleoylphosphatidylglycerol; Avanti Polar lipids) unsized unilamellar liposomes[12] for 30 min at RT. Trypsin (bovine pancreas, 5000 USP/mg, Sigma-Aldrich) was added to a final concentration of 0.01 mg/mL and the mixture was incubated for 60 min at 4 °C. The mixture was sampled periodically and the reaction in each sample was stopped by adding 5× SDS loading buffer (containing 250 mM Tris, 10% SDS (w/v), 0.2% bromophenol blue (w/v), 50% glycerol (w/v), 500 mM DTT) and immediate heating to 95 °C. The samples were analyzed via SDS-PAGE on a 12% acrylamide gel.

**Limited proteolysis**. IM30* in 20 mM HEPES pH 7.6 was incubated with the endoproteinase GluC on ice at protease:protein ratios of 1:10, 1:100 and 1:1000 for 30 or 60 min. The reaction was stopped by the addition of 5× SDS-sample buffer and subsequent heating to 95 °C. Samples were analyzed by SDS-PAGE on a 12% acrylamide gel. A suitable band was cut and analyzed by in-gel digestion followed by MALDI mass fingerprinting[44].

**FRET measurements**. For FRET (Förster resonance energy transfer) measurements, IM30-CFP and IM30-Venus were expressed as described[43] and copurified after mixing cell pellets prior to cell lysis in a ratio of 27% CFP and 63% Venus (w/w). A solution with 0.2 µM of the purified CFP/Venus labeled IM30 rings was incubated with increasing DOPG concentrations (0–1000 µM lipid, unilamellar liposomes) for ~2 h at RT. Fluorescence was measured using a FluoroMax 4 fluorimeter (Horiba Scientific). For FRET measurements, an excitation wavelength of 420 nm (slit width 3 nm) was chosen and spectra were recorded from 440–700 nm (slit width 3 nm). In order to correct for the contribution of liposome light scattering and to detect a change in the relative contribution of CFP and Venus fluorescence due to decreased FRET, a superposition of spectra measured for the individual components in absence of the others was fitted to each spectrum (Eq. (1)) yielding the fractional contribution $f$ for each spectrum, relative to the corresponding reference spectrum.

$$S_{meas} = f_{lip}S_{lip} + f_{cfp}S_{cfp} + f_{ven}S_{ven} \quad (1)$$

The buffer spectrum was subtracted beforehand. In presence of liposomes $f_{cfp}$ tends to increase, while $f_{ven}$ tends to decrease, indicating reduced FRET. Since the overall change is not very large, the trend in the values for $f$ is overlaid by the slight change in the individual apparent quantum yields, as determined by measuring the CFP and Venus fluorescence in absence of the FRET partner, but presence of lipids. Furthermore, variations in the IM30 concentration leads to scattering of the $f$ values. In order to correct for the variations of IM30 concentration, the data are presented as ratio of $f_{ven}/f_{cfp}$ and finally normalized to the value in absence of liposomes. By comparing the resulting curve with the one observed for the controls ($f_{ven}/f_{cfp}$ obtained individually in absence of the FRET partner) the effect of FRET can be distinguished from the effect of changes in quantum yield due to the presence of liposomes. This procedure was performed for three sets of data (each including control and FRET measurements), and the average and standard error calculated for the resulting normalized $f$-ratio.

**CD spectroscopy**. CD spectra of IM30*, IM30_H3b-7, and IM30 WT (0.1 mg/mL) were measured in absence and presence of 2,2,2-trifluoroethanol (TFE, 8 M) using a JASCO-815 CD spectrometer with an MPTC-490S temperature-controlled cell holder. Spectra were measured from 260 to 190 nm (cell length 0.1 cm, 1 nm data pitch, 5 nm bandwidth, 100 nm/min, 1 s data integration time, averaged over 6 accumulations of the same sample). Spectra were smoothed with a Savitzky-Golay filter and the spectra of three samples were averaged. The secondary structure composition was analyzed with BeStSel[45].

The stability of the secondary structure of IM30 WT in 10 mM HEPES or Tris was measured by urea denaturation. The protein was incubated with 0–5 M urea overnight. CD spectra were measured from 200 nm to 260 nm (2 nm bandwidth, 1 s data integration time, 100 nm/min, 9 accumulations per sample). The ellipticity at 222 nm was plotted against the urea concentration and the resulting denaturation curve was normalized between 0 and 1, assuming full denaturation at 5 M urea. Then the data were fitted with a two-state unfolding model:

$$f_D = \frac{F - U}{1 + e^{(c - T_m)/dc}} + U \quad (2)$$

Where $f_D$ is the fraction of denatured protein, $F$ is the folded state, $U$ is the unfolded state, $c$ is the concentration of urea and $T_m$ is the transition midpoint.

The thermal stability of IM30* at increasing NaCl concentrations and of IM30 WT at increasing isopropanol concentrations was determined via CD spectroscopy. During the temperature ramp, CD spectra were measured from 200 to 250 nm (cell length 0.1 cm, 1 nm data pitch, 5 nm bandwidth, 200 nm/min, 1 s data integration

time, averaged over three accumulations of the same sample). The temperature gradient was set to 15–95 °C (2 °C steps, overall heating rate 0.27 °C/min). Spectra were smoothed with a Savitzky-Golay filter. The denaturation curves (ellipticity at 222 nm vs. temperature) from three independent measurements were averaged. The first derivative of the averaged denaturation curves was used to determine the phase transition temperature as the center of the transition peak.

**Hydrogen-deuterium exchange mass spectrometry**. Hydrogen-deuterium exchange mass spectrometry (HDX-MS) was essentially conducted as described previously[46,47]. Sample preparation was aided by a two-arm robotic autosampler (LEAP Technologies). IM30 or IM30* (50 µM) was diluted 10-fold in D$_2$O-containing buffer (20 mM HEPES-Na pH 7.6). After incubating for 10, 95, 1000 or 10,000 s at 25 °C, the reaction was stopped by mixing with an equal volume of pre-dispensed quench buffer (400 mM KH$_2$PO$_4$/H$_3$PO$_4$ + 2 M guanidine-HCl; pH 2.2) kept at 1 °C and 100 µl of the resulting mixture injected into an ACQUITY UPLC M-Class System with HDX Technology[48] (Waters). Undeuterated samples of IM30 and IM30* were generated similarly by 10-fold dilution in H$_2$O-containing buffer. The injected protein samples were washed out of the injection loop with water + 0.1% (v/v) formic acid at 100 µl/min flow rate and guided to a column of immobilized porcine pepsin enabling protein digestion at 12 °C. The resulting peptides were collected for three minutes on a trap column (2 mm × 2 cm) kept at 0.5 °C and filled with POROS 20 R2 material (Thermo Scientific). The trap column was then placed in line with an ACQUITY UPLC BEH C18 1.7 µm 1.0 × 100 mm column (Waters) and the peptides eluted with a gradient of water + 0.1% (v/v) formic acid (eluent A) and acetonitrile + 0.1% (v/v) formic acid (eluent B) at 30 µl/min flow rate as follows: 0–7 min/95–65% A, 7–8 min/65–15% A, 8–10 min/15% A. The peptides were guided to a Synapt G2-Si mass spectrometer (Waters) equipped with an electrospray ionization source and ionized at a capillary temperature of 250 °C and spray voltage of 3.0 kV. Mass spectra were acquired over a range of 50–2000 $m/z$ in HDMS$^E$ (enhanced high definition MS) or HDMS mode for undeuterated and deuterated samples, respectively[49,50]. [Glu1]-Fibrinopeptide B standard (Waters) was utilized for lock mass correction. During separation of the peptides on the C18 column, the pepsin column was washed three times by injecting 80 µl of 0.5 M guanidine hydrochloride in 4% (v/v) acetonitrile. Blank runs (injection of double-distilled water instead of sample) were performed between each sample. All measurements were carried out in triplicate.

Peptides of IM30 and IM30* were identified and evaluated for their deuterium incorporation with softwares ProteinLynx Global SERVER 3.0.1 (PLGS) and DynamX 3.0 (both Waters). Peptides were identified with PLGS from the non-deuterated samples acquired with HDMS$^E$ employing low energy, elevated energy and intensity thresholds of 300, 100 and 1000 counts, respectively and matched using a database containing the amino acid sequences of IM30, IM30*, pepsin and their reversed sequences. Hereby, the search parameters were as follows: Peptide tolerance = automatic; fragment tolerance = automatic; min fragment ion matches per peptide = 1; min fragment ion matches per protein = 7; min peptide matches per protein = 3; maximum hits to return = 20; maximum protein mass = 250,000; primary digest reagent = non-specific; missed cleavages = 0; false discovery rate = 100. For quantification of deuterium incorporation with DynamX, peptides had to fulfill the following criteria: Identification in at least 4 of the 6 non-deuterated samples; minimum intensity of 25,000 counts; maximum length of 25 amino acids; minimum number of products of two; maximum mass error of 25 ppm; retention time tolerance of 0.5 min. All spectra were manually inspected and omitted if necessary, for example, in case of low signal-to-noise ratio or the presence of overlapping peptides disallowing the correct assignment of the isotopic clusters. HDX-MS data can be found in the supplemental dataset[51].

**Nuclear magnetic resonance (NMR) spectroscopy**. The $^1$H NMR spectrum of a 110 µM sample of IM30_H3b-7 in 20 mM HEPES pH 7.6, 100 mM NaCl supplemented with 10% D$_2$O was recorded on an 800 MHz Bruker Avance III HD NMR spectrometer equipped with a triple resonance HCN-cryogenic probe head at 298 K. Suppression of the water signal was achieved by excitation sculpting, using a Bruker standard pulse sequence. The spectrum was processed with Topspin (Bruker, Karlsruhe, Germany).

**SEC coupled small-angle X-ray scattering (SEC-SAXS)**. SAXS experiments were performed at beamline P12 operated by EMBL Hamburg at the PETRA III storage ring (DESY, Hamburg, Germany). SAXS data, $I(q)$ versus $q$, where $q = 4\pi \sin\theta/\lambda$ is the momentum transfer and $2\theta$ is the scattering angle and $\lambda$ the X-ray wavelength ($\lambda = 1.24$ Å; 10 keV), were collected using online size exclusion chromatography with a splitter, directing half of the eluted sample to MALS light detectors as described in ref. [52] and the remaining half to the beamline for SAXS data collection. The protein was heated to 50 °C and subsequently cooled down to room temperature slowly followed by buffer exchange via SEC to 25 mM HEPES (pH 7.5), 125 mM NaCl, 5% glycerol (w/w) and 2 mM TCEP. This treatment appeared to be necessary, as especially lipids, which tend to stick to IM30 proteins even after purification by usual SEC[16], were removed. The structure of the protein was verified by comparing CD-spectra before and after the procedure (Supplementary Fig. 2g). 75 µL of 14.4 mg/mL IM30* were loaded on a Superdex 200 10/300 GL column (GE Healthcare) equilibrated with 25 mM HEPES (pH 7.5), 125 mM NaCl,

5% glycerol (w/w) and 2 mM TCEP at RT. Each run consisted of 50 min of data-collection, with 3000 frames being collected at an exposure time of 1 s. Data were analyzed using the ATSAS software package[53]. The primary 2D-data underwent standard automated processing (radial averaging), and background subtraction was performed using CHROMIXS[54], combined with additional manual and statistical evaluation (e.g., for radiation damage) to produce the final 1D-SAXS profiles presented here. The molecular mass of the particles across the analyzed peak was calculated based on the methods implemented in CHROMIXS. The values presented in this report are averages of both the consensus Bayesian assessment[55] and the SAXSMoW volume correlation[56] approach for calculating the masses. Estimation of the radius of gyration ($R_g$) by Guinier-plot analysis was performed using the *autorg* function of PRIMUS[57]. The first 19 data points at low angles in the Guinier region were excluded from further analysis. GNOM was used for pair distance distribution analysis of the data within a range of $q = 0.0929$–$7.2965 \, \mathrm{nm}^{-1}$, choosing a $D_{max}$ of 26 nm and forcing the $P(r)$ function to 0 at $D_{max}$[58]. Ab initio modeling via the generation of dummy residue models was performed with GASBOR based on the $P(r)$ function in reciprocal space[59]. The number of dummy residues was set to 290 for a p2 particle symmetry. A p2 symmetry was assumed, as choosing higher degrees of freedom did result in bead models with higher $\chi^2$ values. 115 GASBOR bead models were generated in total. The bead models were clustered by running DAMCLUST and setting a p2 symmetry and considering only backbone atoms to ignore water molecules in the GASBOR models[60].

**Model building**. IM30 dimer models were generated according to the scheme presented in Supplementary Fig. 11b. From the clusters generated by DAMCLUST, one isolated cluster (cluster 14) was excluded from further analysis. For each of the other clusters, the most typical bead model according to DAMCLUST was chosen. Water molecules in the bead model were removed. Then the model was transformed into a density map with a resolution of 4 Å by the Molmap command implemented in CHIMERA[61]. A resolution of 4 Å was chosen because the beads were treated as hard spheres and have a diameter of 3.8 Å. The resulting dimer maps were split along the symmetry axes to create maps of the monomer using SEGGER[62]. To fit IM30 into the map, a predicted structure of the monomer (IM30_Saur2017[19]) was used as an initial template. The predicted structure was fragmented by removing the loops and keeping the helices intact, yielding six helix fragments (Supplementary Fig. 11a). The fragments were placed manually into the map to roughly fit the density. MODELLER was used to recreate the missing loop regions between the fragments and to remodel the parts of the structure, which are considered to be flexible according to the results of the HDX measurements of IM30* [63]. A threshold of 45% relative HDX (after 10 s) was set as the limit to define a part of the structure as flexible. We refined the models by a simulated annealing molecular dynamics (MD) approach guided by the density map using FLEX-EM[64,65]. At least 50 runs of subsequent MD refinement were performed, using a cap shift of 0.15 to restrain secondary structure elements and keeping helix 2 and 3a as rigid bodies. Two of each refined monomer structures were then placed into the dimer maps by exhaustive One-At-A-Time 6D search (colores) and simultaneous multi-fragment refinement (collage), using the SITUS package[66]. Where necessary, clashes in the dimer interfaces were removed by running a short minimization procedure implemented in CHIMERA (100 steepest descent steps, step size 0.02 Å, 10 conjugate gradient steps, step size 0.02 Å)[67,68].

**ACMA proton leakage assay**. An aliquot of unsized unilamellar DOPG liposomes (400 µM lipid concentration, in 20 mM HEPES pH 7.6 + 150 mM KCl) was incubated with 2.4 µM protein for 5 min at RT. Thereafter, the mixture was diluted with 20 mM HEPES pH 7.5 + 150 mM NaCl and isopropanol to a final concentration of 6% isopropanol (v/v), 100 µM lipid and, 0.6 µM protein (Noteworthy, the secondary structure and overall stability of IM30 were preserved at 6% isopropanol (Supplementary Fig. 6)). 1 µL ACMA (9-amino-6-chloro-2-methoxyacridine) was added to a final concentration of 2 µM. The sample was then incubated for another 200 s at RT in a 4 mL glass cuvette with continuous stirring. The fluorescence intensity was measured with a FluoroMax 4 fluorimeter (Horiba Scientific), using an excitation wavelength of 410 nm (slit width 2 nm), an emission wavelength of 490 nm (slit width 2 nm) and a measurement interval of 0.1 s. The measurement was started by addition of 1 µL valinomycin (final concentration 0.02 µM), to render the liposomes permeable for K$^+$, which results in formation of a proton gradient across the membrane. The fluorescence intensity was monitored for 300 s with continuous stirring. Thereafter, the proton gradient was quenched by the addition of CCCP ([(3-chlorophenyl)hydrazono]malononitrile) to a final concentration of 2 µM, and the fluorescence intensity was monitored for another 100 s.

The fluorescence intensity was normalized by setting the intensity to 100% prior to the addition of valinomycin and the intensity 100 s after the addition of CCCP to 0%. The initial slopes were estimated by a linear fit over 10 to 30 s after addition of valinomycin.

**Laurdan fluorescence measurement**. Unsized unilamellar DOPG liposomes containing Laurdan (6-dodecanoyl-N, N-dimethyl-2-naphthylamine, from Sigma, Taufkirchen, Germany) (molar ratio lipid:Laurdan = 1:500) were produced as

described elsewhere[12]. To analyze the kinetics of IM30 membrane binding, liposomes and protein were mixed to a final concentration of 2.5 µM IM30 and 100 µM lipid. Fluorescence emission spectra were recorded at 25 °C over 20 min every 20 s using a FluoroMax-4 spectrometer (Horiba Scientific) from 425 to 505 nm upon excitation at 350 nm The excitation and emission slit width was set at 1 nm and 10 nm, respectively. The generalized polarization (GP) defined by Parasassi et al.[69] was calculated according to Eq. (3). ΔGP values were calculated via subtraction of the linear fit function of the DOPG control from the measurements in presence of protein.

$$\mathrm{GP} = \frac{I_{440} - I_{490}}{I_{440} + I_{490}} \tag{3}$$

**Atomic force microscopy (AFM)**. To visualize IM30-binding to mica surfaces, 50 µL adsorption buffer (10 mM Tris, 150 mM KCl, 25 mM MgCl$_2$, pH 7.6 or 10 mM HEPES, 150 mM KCl, 25 mM MgCl$_2$, pH 7.6) was incubated on freshly cleaved muscovite mica (12 mm diameter; Ted Pella Inc. grade V1) for 5 min at RT. All buffers and solutions were freshly prepared and filter sterilized (0.2 µm filter) before use. The mica substrate was washed two times with 50 µL of adsorption buffer. Then, 5 µL IM30 WT was added to a final concentration of ~0.5 µM. The protein was incubated on the substrate for 10 min at RT. Thereafter, the substrate was washed with ~1 mL imaging buffer (10 mM Tris, 150 mM KCl, pH 7.6 or 10 mM HEPES, 150 mM KCl, pH 7.6).

To visualize IM30 binding on membranes, a solid-supported lipid bilayer (SLB) was prepared as follows: a freshly cleaved muscovite mica disc (12 mm diameter; Ted Pella Inc. grade V1) was washed with adsorption buffer (20 mM MgCl$_2$, 20 mM HEPES, pH 7.6) two times (50 µL). All buffers and solutions were freshly prepared and filter sterilized (0.2 µm filter) before use. 50 µL of the adsorption buffer was left on the mica, and 50 µL liposome suspension (100% DOPG or 40% DOPG 60% DOPC, 5 mg/mL unilamellar liposomes[12]) was added. The solution on the mica disc was gently mixed by pipetting a volume of 50 µL up and down two to three times. Then, the mixture was incubated on the mica disc for 20–30 min at RT. Afterward, the mica was washed with 1 mL imaging buffer (20 mM HEPES pH 7.6), and a drop of 100 µL buffer was left on the mica disc.

The samples were imaged with a Nanowizard IV AFM (JPK) using uncoated silicon cantilevers (OMCL AC240; Olympus, tip radius 7 nm, resonance frequency ~70 kHz and ~2 N/m spring constant). Measurements were carried out in QI mode or tapping mode in imaging buffer at ~30 °C. The force setpoint was set as low as possible, typically around 5 nN for measurements on SLBs, and <1 nN for measurements on mica. Formation of an intact lipid bilayer was confirmed by analysis of force-distance curves with high setpoint[70] and by imaging the bilayer before protein addition. The protein was added to the sample in small volumes (30–50 µL) to achieve a final solution of roughly 1.5 µM. Images were scanned with 512 × 512 px or 256 × 256 px and 4.8 ms (or 6 ms) pixel time. The resulting images were analyzed with GWYDDION[71]. The measured height-images were leveled by removing a polynomial background, and scan rows were aligned by fitting a second-degree polynomial and aligning the offsets of the substrate or the lipid surface. The images were cropped to the area of interest. Full images are shown in Supplementary Fig. 5a, b.

**Reporting summary**. Further information on research design is available in the Nature Research Reporting Summary linked to this article.

## Data availability
The authors declare that the data supporting the findings of this study are available within the paper and its Supplementary information files. The data used to generate graphs and charts shown in Figs. 1a, b, d, e, 2a–f are provided in the Supplementary Data 1. The HDX-MS data used to generate Fig. 3b are provided in the Supplementary Data 2. All other relevant data are available from the corresponding author upon reasonable request.

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

## Acknowledgements

This work was funded by the Max-Planck graduate center at the Max-Planck institutes and the University of Mainz. We thank Prof. Dr. Elmar Jaenicke (Institute for Molecular Physiology, Johannes Gutenberg University Mainz) assistance with MALS measurements, Mario Dejung (Institute of Molecular Biology, Johannes Gutenberg University) for assistance with MS analysis and Sabine Wecklein (Department of Chemistry - Biochemistry, Johannes Gutenberg University Mainz) for support with the data acquisition of ACMA proton leakage measurements. The synchrotron SAXS data were collected at beamline P12 operated by EMBL Hamburg at the PETRA III storage ring (DESY, Hamburg, Germany). We would like to thank the beamline staff (especially Dr. Nelly Hajizadeh) for assistance in using the beamline. We thank the Deutsche Forschungsgemeinschaft for co-financing the HDX-system and support through the "DFG core facility for interactions, dynamics and macromolecular assembly structure" at the Philipps-University Marburg. Molecular graphics and analyses were performed with UCSF Chimera, developed by the Resource for Biocomputing, Visualization, and Informatics at the University of California, San Francisco, with support from NIH P41-GM103311. We thank Prof. Dr. Luning Liu and Dr. Longsheng Zhao (University of Liverpool, Institute of Integrative Biology) helping us find the necessary imaging parameters and conditions for atomic force microscopy of IM30 rings on mica.

## Author contributions

B.J., C.S., J.H., U.H., W.S., N.H., and D.S. were responsible for the general protein characterization study conception. The AFM studies were conceptualized by B.J., A.A., and S.W. The SAXS studies were conceptualized by B.J., R.O., and E.W. B.J., C.S., and J.H. prepared the protein and liposome samples. B.J., C.S., J.H., and N.H. collected and analyzed the general protein characterization data (fluorescence spectroscopy, CD spectroscopy, SEC, SEC-MALS, gel electrophoresis). B.J. and A.A. collected and analyzed the AFM data. R.O. collected the SAXS data. B.J. and R.O. analyzed the SAXS data. U.H. collected and analyzed the NMR data. W.S. collected and analyzed the HDX data. B.J. built and analyzed the structural models. B.J., A.A., C.S., and W.S. visualized the data. W.S., U.H., E.W., S.W., and D.S. were responsible for supervision, project administration, funding acquisition, and resources.

## Funding

## Competing interests

The authors declare no competing interests.
