## [Peer Review File · Communications Biology]

Reviewers' comments:

Reviewer #1 (Remarks to the Author):

VIPP1 is a IM30 family protein of plastids and cyanobacteria that has a distinctive C-terminal domain in addition to the domains found in other IM30 family proteins. VIPP1 is implicated in thylakoid biogenesis and membrane stabilization under stress conditions. In earlier studies, it has been shown by the authors that VIPP1 can form ring structures, and the authors have also demonstrated that these rings can associate with liposome surfaces, and that smaller associations can associate even better with membranes. In their manuscript, Junglas et al. now describe a flat association of VIPP1 protomers on membrane surfaces, which could be the basis for a membrane stabilization by this protein. In the following, I will describe and comment on the findings. Questions are indicated by numbers.

The authors first observed a partial protection of trypsin cleavage sites in VIPP1 by the presence of DOPG vesicles, which are highly charged. They conclude that the protein must interact with membranes in a way that results in the observed protection. Then the authors used CFP- and VENUS-labeled VIPP1 subunits that were randomly mixed with an excess of VENUS-labeled protein for FRET analyses. They observed a slight reduction in FRET intensity that indicates an increased distance of still associated VIPP1 protomers at the membrane surface. They mention in the methods that the individual fluorophores per se showed also a FRET-like behaviour induced by lipids without any FRET, i.e. a lipid induced decrease of donor emission in the absence of the acceptor and a lipid induced increase of acceptor emission in the absence of a donor, which was not taken into account due to its low (<5%) effect. However, as FRET was measured ratiometrically, i.e. by determining the fraction of the excited donor whose energy is transferred to the acceptor, the maximum FRET difference was about 5%, decreasing from 0.54 to 0.49 (Fig. 1A), which could be exactly the up to 5% effect of the dyes alone without FRET partner.

(1) It is therefore not clear to me, whether the shown non-corrected FRET data are really FRET signals or rather lipid effects on the respective donor or acceptor dyes. I am sure that the authors can clarify this confusion.

Then the authors analyzed surface and membrane interactions by atomic force microscopy. These are nice data. VIPP1 formed carpet-like covered areas on PG bilayers. A variant that formed only dimers in solution gave similar but significantly thicker layers.

(2) The authors might comment on that aspect: what could have caused the distinct thicknesses? The authors then include an important functional aspect by demonstrating that PG vesicles could indeed be stabilized by VIPP1. In a smart fluorescence-based assay that relies on the action of potassium-specific transport of valinomycin out of vesicles that can only be compensated by proton import through the membrane bilayer, they showed that this proton leakage was largely suppressed by VIPP1, the dimeric form that previously gave thicker „carpets“ being even a stronger-acting variant than the wild type.

(3) The authors should include the CCCP-induced regain of ACMA fluorescence in that Figure 1d. Another assay for lipid binding was then added, that confirmed the above observed interaction with PG liposomes. As the dimeric mutated variant of VIPP1 showed faster membrane binding and better stabilization activity, the authors continued to characterize this protein in comparison to wild type protein by small-angle light scattering and CD-analyses. It appears that the dimeric form has larger unstructured regions than the wild type. Limited proteolysis experiments confirmed that the N-terminal helix 1 and regions from helix 4 to 7 are likely less structured. Then the authors attempted to model the unstructured regions of that protein, using the known X-ray structure of the region up to helix 3 from the related bacterial PspA as fixed starting structure and SAXS data, fitting possible

structures into the given SAXS envelopes. The data were suggestive for dimerization mediated by the C-terminal unstructured domain. Single Cys substitutions were generated that resulted in disulfides, confirming the interaction of the predicted regions.

The manuscript is well written and contains novel aspect that are definitively worth to be published. Besides the above mentioned questions, there are only few more points that I would suggest to discuss:

(4) Are the carpet formations and membrane-stabilizing effects restricted to pure PG bilayer interactions? Thylakoids have only ca. 13% PG. Instead, most of the natural lipids in these membranes are uncharged galactolipids. There is the risk that the observations are induced by these extremely charged membrane surfaces.

(5) The authors might think of citing in the context of the potential stabilizing mechanism a PspA scaffold paper that stated in 2008: „Such large scaffolds could support membrane integrity even better than a ring could do and promote the closure of transient holes or leaks by preventing their extension. Multiple membrane interactions of the PspA scaffold could stabilize a flat conformation of the scaffold at the membrane, which then could cover large membrane regions.“ (Standar et al. 2008, FEBS Letters 582, 3585–3589)

To conclude, the manuscript is successfully integrating a very broad range of biochemical and biophysical methods to show that VIPP1 can stabilize PG vesicles in form of flat protein associations that are generated from re-structuring of ring shaped soluble associations. This mechanism may well play a role in the natural membrane system and therefore the publication of this manuscript should be published to stimulate more research in this field.

Reviewer #2 (Remarks to the Author):

In this paper, the authors present two main results:

1. IM30 oligomers disassemble upon binding to negatively charged surfaces (DOPG membranes) and protect lipid membranes against isopropanol dissociation.
2. The C-terminal end of a new IM30* mutant is intrinsically disordered and dimerizes.

The authors present a wealth of experimental data, considering the short length of the nature communication format, the manuscript is very dense.

A- First, the authors present limited proteolysis with trypsin which show that IM30 is protected in the presence of DOPG liposome. In the main manuscript, the author should mention clearly the increased protection offered by the membrane. Instead of structural rearrangement, this effect could be due shielding of the helix H1-6 by the membrane.

Also, a more detailed localization of the membrane-associating regions could be inferred from further limited proteolysis experiment on IM30 associated with liposomes (as they performed using endoproteinase GluC latter).

B- Then, the author present FRET data on WT IM30, to show that IM30 undergoes a conformational change upon DOPG binding. They used constructs that they have already used in ref 43.

They explain line 46: “This FRET change remained small and leveled off at high lipid concentrations”. I have one question relating to this sentence:

-The CFP-IM30 / Venus-IM30 were incubated with liposomes only two hours (120min). But the Laurdan experiment they performed indicates that the equilibrium is only reached after more than 800 min. Maybe the author should allow the IM30-liposomes binding equilibrium to be reached before performing the FRET measurement.

B- Then, the authors performed AFM on IM30. In the supplementary material, they show really convincing data using TRIS, that are consistent with their TEM model (ref 19). Over-time this ring structure seems to disassemble, but in a long enough time to allow AFM data to be collected. The AFM maps they collected using HEPES was of lower quality and showed very fast disassembly of the IM30 ring.

I guess the negatively charged mica surface is problematic and disassemble IM30 ring, and TRIS could act as a protecting factor.

Contrary to this limitation, the author chose to use HEPES to record AFM maps of IM30 with or without DOPG. I do not agree with the argument that in solution, the unfolding mid-point for IM30 shifts from 2.7 to 2.4 M UREA when using HEPES instead of TRIS, as this does not take into account the mica surface. The increased coiled-coil content found using TRIS might be the relevant conformation for ring formation.

In conclusion, I think the bad quality of the reference AFM maps (IM30 in isolated) using HEPES, that show 'shadowing' of the ring structure, does not allow a good reference for the interpretation of the 'carpet-like' feature upon addition of DOPG.

Can the author record the AFM-liposome AFM images with TRIS buffer?

C- The authors introduce a mutant with mutation in helix 4. IM30-FERM mutant was already analyzed by the same authors in ref 19 and was shown to oligomerized as a tetramer (and do not form ring). Here, they introduce two new mutations in loop 2 E83A, E84A, which shift the oligomerization to a dimer (NaCl dependent).

As they have shown in a previous study (ref 27) this new IM30* construct has a higher affinity for membrane (Laurdan experiment). Line 77: "This indicates that membrane binding of IM30 WT rings is less effective than binding of smaller IM30* oligomers". The correct wording should be : "... of IM30 WT rings is slower than binding...". Since the equilibrium is not reached at the end of the experiment, we cannot conclude for the situation at equilibrium.

Here, the authors show for the first time that enhancing the membrane affinity by the mutation increases the membrane protection against isopropanol dissociation.

The AFM maps of IM30*-DOPG also show marked difference compared to IM30-DOPG. The conclusion that IM30* forms a carpet specifically in the presence of DOPG should be confirmed with a control of IM30* alone in isolation. This control is lacking. This is required even-more since the control of IM30 alone with HEPES shows disassembly of the rings.

The rest of the manuscript is describing the structural feature of IM30* dimer. I wonder of the relevance of this state in the context of the manuscript. The above AFM (and FRET) data show that IM30 and IM30* disassemble on the membrane. In this case, the isolated oligomer is not the functional state. Any structure of the soluble oligomer cannot reveal how IM30* protect the membrane. A structure of IM30* bound to membrane would be more relevant (although much

more challenging to obtain). I acknowledge how difficult this could be, but I would suggest solid-state NMR. But again, localizing further the regions of IM30 in interaction with the membrane (limited proteolysis) would be an original result.

The SAXS, CD, HDX and limited proteolysis on the full length IM30* all converge to conclude that helices 3b-7 is disordered. Nevertheless, this mutant has a quadruple mutation in helix 4, and these mutations are likely to destabilize an important folding core (in particular the F to A mutation). The fact that the mutant does not form large ring underpins the importance of this missing fold. Here, the disorder could well be an experimental artifact, and is not worth deep investigation such as extended modeling.

Line 108-109: The authors write: "The major structural difference between IM30* and IM30 WT lies in the region of the predicted helices 1, 3b, 4, and 5/6", which for me indicates that these regions are NOT involved in membrane binding, but more likely in ring formation.

On the contrary, the fact that the mutant keeps a high affinity for membranes suggest that the activity is localized at the H2-3a coiled-coil. Coiled-coil regions are known to have affinity for membrane.

The NMR data on the truncated construct cannot be conclusive unless the author validate the 'divide and conquer' approach by showing that the NMR spectrum of the full-length oligomer superimposed with that of H2-3b + H1 + H3b-7. Truncation of H3b-7 can lead to unfolding of this truncated product.

From the modelling refined with the SAXS data, the authors conclude that the C-terminal tail is disordered but is the dimeric interface. There is an apparent contradiction here, and to-date, only few example of peptides are involved in an interaction interface while remaining disordered (absence of folding upon binding, for example the study by Borgia et al, Nature, 2018, 555 p 61-68 on the histone H1 tail). To be convincing, other experimental data should be added, such as NMR (chemical shift and PRE-NMR), FRET, fluorescent anisotropy, etc...

Also, the addition of cysteine before SDS-PAGE cannot be conclusive for the presence of dimer in the native state. Upon SDS addition and denaturation, any cysteine is available for disulfide-bound formation, whether it was in a folded domain or not in the native state. Native mass spectrometry would be more conclusive.

Finally, the authors mention liquid-liquid phase separation and membrane-less organelles in their conclusion. This seems out of the topic of this study, as IM30 is a membrane-associating protein.

Response to reviewers

We appreciate the reviewers' constructive remarks that have enabled us to improve our manuscript "*IM30 IDPs form a membrane protective carpet upon supercomplex disassembly*". Below, the reviewers' remarks are reproduced in *italics*; our reply and the resulting changes to the manuscript appear in regular font.

Reviewer #1:

Comment 1:

The authors first observed a partial protection of trypsin cleavage sites in VIPP1 by the presence of DOPG vesicles, which are highly charged. They conclude that the protein must interact with membranes in a way that results in the observed protection. Then the authors used CFP- and VENUS-labeled VIPP1 subunits that were randomly mixed with an excess of VENUS-labeled protein for FRET analyses. They observed a slight reduction in FRET intensity that indicates an increased distance of still associated VIPP1 protomers at the membrane surface. They mention in the methods that the individual fluorophores per se showed also a FRET-like behaviour induced by lipids without any FRET, i.e. a lipid induced decrease of donor emission in the absence of the acceptor and a lipid induced increase of acceptor emission in the absence of a donor, which was not taken into account due to its low (<5%) effect. However, as FRET was measured ratiometrically, i.e. by determining the fraction of the excited donor whose energy is transferred to the acceptor, the maximum FRET difference was about 5%, decreasing from 0.54 to 0.49 (Fig. 1A), which could be exactly the up to 5% effect of the dyes alone without FRET partner.

It is therefore not clear to me, whether the shown non-corrected FRET data are really FRET signals or rather lipid effects on the respective donor or acceptor dyes. I am sure that the authors can clarify this confusion.

Reply: We entirely agree with this reviewer to have to consider the lipid effects. Therefore, as suggested, we re-analyzed the data (as now described in the methods sections), and have now determined the fraction of CFP and Venus fluorescence in the FRET spectra via fitting the corresponding individual spectra to the measured sum-spectrum. If the FRET efficiency changes, the fractional contribution of the CFP and Venus fluorescence should change in a synchronous manner. In order to reduce the impact of slightly different concentrations of the protein, which directly change the fractional contribution, the data are presented as the ratio of the fractions of Venus and CFP. The same procedure was performed with the control experiments (Venus and CFP fluorescence in dependence on lipid concentration, but absence of the FRET partner). This now allows to clearly separate the effect of FRET changes from "background" effects. We now show the data in a new Figure 1a.

Comment 2:

Then the authors analyzed surface and membrane interactions by atomic force microscopy. These are nice data. VIPP1 formed carpet-like covered areas on PG bilayers. A variant that formed only dimers in solution gave similar but significantly thicker layers.

The authors might comment on that aspect: what could have caused the distinct thicknesses?

Reply: The accuracy in the determination of sample thickness is limited in AFM, since the structures under investigation are very soft. The height is determined by the tip position at the maximum loading force during a QI force-distance cycle. Although we chose the lowest possible loading force to minimize sample deformation, there will be an effect of sample deformation that leads to an error in the height that we estimate to be 1 nm. We re-analyzed our measurements and found the sample thickness varying in the range of 0.7 – 1.9 nm for both proteins. We have added this information to the manuscript on page 5, line 62-64 and in the legend of Figure 1. Thus, while it is possible that the thickness of the carpets formed from the wt protein differs from the ones formed from the mutant, we do refrain from further discussing this, as our data do not allow to safely identify differences below 1 nm due to sample restraints.

Comment 3:

The authors then include an important functional aspect by demonstrating that PG vesicles could indeed be stabilized by VIPP1. In a smart fluorescence-based assay that relies on the action of potassium-specific transport of valinomycin out of vesicles that can only be compensated by proton import through the membrane bilayer, they showed that this proton leakage was largely suppressed by VIPP1, the dimeric form that previously gave thicker „carpets“ being even a stronger-acting variant than the wild type.

The authors should include the CCCP-induced regain of ACMA fluorescence in that Figure 1d.

Reply: We apologize for not having properly explained in the manuscript that addition of CCCP does not result in “regain of ACMA” fluorescence but in completely abolishing the delta pH resulting in minimal ACMA fluorescence. This value was set as “0”. We now write on page 22, line 436-438: “Thereafter, the proton gradient was quenched by the addition of CCCP ([3-chlorophenyl)hydrazono]malononitrile) to a final concentration of 2 μ M, ...”

Comment 4:

[...] The manuscript is well written and contains novel aspect that are definitively worth to be published. Besides the above mentioned questions, there are only few more points that I would suggest to discuss:

Are the carpet formations and membrane-stabilizing effects restricted to pure PG bilayer interactions? Thylakoids have only ca. 13% PG. Instead, most of the natural lipids in these membranes are uncharged galactolipids. There is the risk that the observations are induced by these extremely charged membrane surfaces.

Reply: In fact, the content of negatively charged lipids in cyanobacterial and thylakoid membranes is higher, when both negatively charged lipid species, PG and SQDG, are considered, but typically does not exceed 40% (see e.g. Sakurai I, et al. (2006) J Biochem 140: 201–209). Yet, this difference in overall surface charge needs to be considered. To exclude that the observed disassembly and carpet formation are induced by the extreme surface charge, we have repeated the AFM experiments with a PC:PG (60:40) mixture (Supplementary Fig. 5 c and d). On these surfaces we also see ring disassembly and formation of carpet structures. We would like to note that we were unable to perform the liposome stability assay using this lipid composition, as at this composition the liposomes were not sufficiently stable.

Comment 5:

The authors might think of citing in the context of the potential stabilizing mechanism a PspA scaffold paper that stated in 2008: „Such large scaffolds could support membrane integrity even better than a ring could do and promote the closure of transient holes or leaks by preventing their extension. Multiple membrane interactions of the PspA scaffold could stabilize a flat conformation of the scaffold at the membrane, which then could cover large membrane regions.“ (Standar et al. 2008, FEBS Letters 582, 3585–3589)

Reply: We thank the referee for mentioning this important reference. We unintentionally removed the reference in the final phase when shortening the manuscript.

We completely agree that this is an important reference, as it states for the first time (to the best of our knowledge) that members of the PspA family can assemble in “scaffold” structures. We have re-added the suggested reference and rephrased the corresponding paragraph. Now we write on page 12, line 229-233: “...In contrast, IM30 carpets apparently suppress proton leakage in liposomes and thereby maintain the integrity of membranes, as previously suggested for its ancestor PspA, which is thought to form scaffold-like structures to cover large membrane areas and prevent leakage^{38,39}. The idea of IM30 and PspA having similar membrane stabilizing functions is in agreement with the observation that IM30 can functionally complement *E. coli pspA* null mutants⁴⁰.”

Comment 6:

To conclude, the manuscript is successfully integrating a very broad range of biochemical and biophysical methods to show that VIPP1 can stabilize PG vesicles in form of flat protein associations that are generated from re-structuring of ring shaped soluble associations. This mechanism may well play a role in the natural membrane system and therefore the publication of this manuscript should be published to stimulate more research in this field.

Reply: We thank the reviewer for the positive assessment.

Reviewer #2:

Comment 1:

The authors present a wealth of experimental data, considering the short length of the nature communication format, the manuscript is very dense.

Reply: We thank this referee for acknowledging the huge amount of data (and work) incorporated and presented in the current manuscript.

Comment 2:

[...] First, the authors present limited proteolysis with trypsin which show that IM30 is protected in the presence of DOPG liposome. In the main manuscript, the author should mention clearly the increased protection offered by the membrane. Instead of structural rearrangement, this effect could be due shielding of the helix H1-6 by the membrane. Also, a more detailed localization of the membrane-associating regions could be inferred from further limited proteolysis experiment on IM30 associated with liposomes (as they performed using endoproteinase GluC latter).

Reply: We thank the reviewer for commenting on the unprecise description in the main text. We reworded the paragraph accordingly and now write in the main text on page 4, line 47-51:

“Supporting the hypothesis that IM30 rings undergo a structural rearrangement upon membrane binding, we observed differences in the trypsin-digestion pattern of IM30 in absence vs. presence of phosphatidylglycerol (PG)-containing liposomes (Supplementary Fig. 1). Yet, these observations do not allow to clearly discriminate between rearrangements of the IM30 structure, shielding of IM30 regions due to membrane binding, or a combination of both.”

We agree that identifying the membrane-associating regions, or better: the structure of the membrane-bound protein, would be highly interesting. Based on previous experimental results, we already know that the isolated, helix 2/3-containing coiled-coil region binds to membrane surfaces. We now add this important information to the discussion on page 11, line 204-205 and write: “This observation is perfectly in line with the recent notion that the isolated helix 2-3 coiled-coil effectively binds to membrane surfaces³⁰.”

However, encouraged by this reviewer, we started to analyze altered protection of the IM30 regions in the presence vs. absence of membrane surfaces. Unfortunately, the picture turned out to be rather complicated, as not only membrane binding but also protein-protein interactions resulting in carpet formation appear to change the accessibility of protein regions. Identification of the membrane-bound IM30 structure is currently a major project that will be addressed by our group in the coming years.

Comment 3:

Then, the author present FRET data on WT IM30, to show that IM30 undergoes a conformational change upon DOPG binding. They used constructs that they have already used in ref 43.

They explain line 46: “This FRET change remained small and leveled off at high lipid concentrations”. I have one question relating to this sentence: -The CFP-IM30 / Venus-IM30 were incubated with liposomes only two hours (120min). But the Laurdan experiment they performed indicates that the equilibrium is only reached after more than 800 min. Maybe the author should allow the IM30-liposomes binding equilibrium to be reached before performing the FRET measurement.

Reply: We agree entirely with this reviewer that it is generally desirable to perform such experiments under equilibrium conditions.

In the original manuscript version we had shown that after a rapid initial increase in the FRET signal, the signal continuously increased slowly to a minor extent. This observation was somewhat puzzling for us, and inspired by this reviewer’s comment, we started to reevaluate our data. While we do not understand the physical basis, we observed that the Laurdan signal alone i.e., the signal coming from Laurdan-labelled liposomes in absence of protein, steadily increased (see Figure below). This increase de facto explains why we did not observe a steady state even after several hours.

We have therefore repeated the entire measurement applying different measurement parameters and now show the new data in Figure 1e. As one can see, the important information gained from this experiment + figure is still the same, as the mutant protein binds faster to membrane surfaces than the ring-forming wt protein. However, equilibrium is reached already after less than 10 min for both wt and the mutant. Thus, as the sample was incubated for 120 minutes for the FRET measurement, these measurements were performed under equilibrium conditions.

Figure: The Laurdan GP value constantly increases even in absence of protein.

Comment 4:

Then, the authors performed AFM on IM30. In the supplementary material, they show really convincing data using TRIS, that are consistent with their TEM model (ref 19). Over-time this ring structure seems to disassemble, but in a long enough time to allow AFM data to be collected. The AFM maps they collected using HEPES was of lower quality and showed very fast disassembly of the IM30 ring.

I guess the negatively charged mica surface is problematic and disassemble IM30 ring, and TRIS could act as a protecting factor.

Contrary to this limitation, the author chose to use HEPES to record AFM maps of IM30 with or without DOPG. I do not agree with the argument that in solution, the unfolding mid-point for IM30 shifts from 2.7 to 2.4 M UREA when using HEPES instead of TRIS, as this does not take into account the mica surface. The increased coiled-coil content found using TRIS might be the relevant conformation for ring formation.

Reply: Assuming a protective “activity” of TRIS is definitely feasible. Actually, in previous experiments we have already found that the protein is not able to fulfill most of the described (*in vitro*) activities anymore in the presence of TRIS. Clearly, this reviewer could not be aware of these (unpublished) details. Most importantly: in the presence of TRIS, IM30 does not bind to membrane surfaces anymore (see also our reply to Comment 4 below). As membrane binding is *the one* IM30 feature that is best established *in vitro* as well as *in vivo*, TRIS clearly does not have a protective activity, instead it appears to inhibit significant conformational changes relevant for the protein activity. Very likely, TRIS “locks” the oligomeric protein in a non-native, inactive conformation or hinders required structural rearrangements. Actually, this further supports our notion of the ring undergoing conformational changes upon membrane binding.

We agree that our observation concerning the stability of the soluble ring in presence of TRIS not necessarily also implies the existence of a more stable ring structure on the membrane since the effect of membrane binding might overrule the stabilizing effect. However, the results themselves clearly show TRIS stabilizing the oligomeric ring both in solution and on the membrane.

We merely used the experiment to make the point that IM30 rings can in principle be visualized using AFM (while knowing that TRIS alters the characteristics of the protein in solution).

Comment 5:

In conclusion, I think the bad quality of the reference AFM maps (IM30 in isolated) using HEPES, that show ‘shadowing’ of the ring structure, does not allow a good reference for the interpretation of the ‘carpet-like’ feature upon addition of DOPG.

Can the author record the AFM-liposome AFM images with TRIS buffer?

Reply: Unfortunately, the referee’s remark: “*bad quality*” *AFM maps* is unclear in our view. Our provided AFM images are as “raw” and unprocessed as possible to avoid any unwanted changes to the surface structures or processing artefacts. We could have achieved “nicer” images by applying in-depth image processing to the data. We also would like to stress that in contrast to most publications on membrane proteins, IM30 attaches to the membrane and is not embedded into the membrane. If proteins are at least partially embedded into the membrane, imaging conditions will be more stable due to the spatial fixation of the protein structure. Nevertheless, all presented data support our claims, i.e., that IM30 rings disassemble on mica surfaces (but do not form carpets, as they do on a membrane surface). To further clarify this, we write in the Supplement on page 6, line 114-118:

“Nevertheless, even samples that were incubated for longer times on mica without imaging showed the additional material around/below them in the first scan (Supplementary Fig. 3b). Consequently, IM30 rings appear to disassemble upon interaction with the negatively charged

mica surface. However, in contrast to the interaction with DOPG bilayers, no carpet formation was observed.”

As requested by the reviewer, we have now tried to obtain structural information about IM30 bound to DOPG membranes in the presence of TRIS buffer. Noteworthy, as already mentioned above, in previous experiments we saw that IM30 does not bind to membranes in the presence of TRIS when binding is tested in a sucrose gradient. In line with these results, we did not observe IM30 on the membrane surface in our AFM experiments (see images below).

Figure: AFM analysis of IM30-binding to DOPG bilayer surface measured in TRIS buffer.

No binding of IM30 to DOPG bilayers in TRIS buffer was observed in AFM images.

Thus, these results further suggest that IM30 membrane binding involves (or even requires) ring disassembly, and the interaction of TRIS with IM30 appears to lock the ring in a binding-incompetent conformation.

Comment 6:

The authors introduce a mutant with mutation in helix 4. IM30-FERM mutant was already analyzed by the same authors in ref 19 and was shown to oligomerized as a tetramer (and do not form ring). Here, they introduce two new mutations in loop 2 E83A, E84A, which shift the oligomerization to a dimer (NaCl dependent).

As they have shown in a previous study (ref 27) this new IM30 construct has a higher affinity for membrane (Laurdan experiment). Line 77: “This indicates that membrane binding of IM30 WT rings is less effective than binding of smaller IM30* oligomers”. The correct wording should be : “... of IM30 WT rings is slower than binding...”. Since the equilibrium is not reached at the end of the experiment, we cannot conclude for the situation at equilibrium.*

Reply: We thank the reviewer for indicating the misleading wording in our manuscript. We rephrased the paragraph accordingly. As suggested, we now write on page 6, line 95-96: “This indicates that membrane binding of IM30 WT rings is slower than the binding of smaller IM30* oligomers.”

Comment 7:

Here, the authors show for the first time that enhancing the membrane affinity by the mutation increases the membrane protection against isopropanol dissociation.

The AFM maps of IM30-DOPG also show marked difference compared to IM30-DOPG. The conclusion that IM30* forms a carpet specifically in the presence of DOPG should be confirmed with a control of IM30* alone in isolation. This control is lacking. This is required even-more since the control of IM30 alone with HEPES shows disassembly of the rings.*

Reply: We thank this referee for requesting this important control, which we now have performed. As visible in the AFM image shown below, IM30* alone in HEPES puffer does not form carpet structures on mica surfaces in absence of a membrane. We thus describe this observation now on page 5, line 71-73 and write: “Noteworthy, carpet formation was not observed when IM30 WT or IM30* were incubated on mica surfaces, i.e. in the absence of a membrane (data not shown).”

Figure: AFM analysis of IM30* on mica surfaces in HEPES buffer.

AFM images of IM30* on mica in HEPES buffer show binding of the protein to the surface as small diffuse particles. No formation of carpet-like structures was observed.

Comment 8:

The rest of the manuscript is describing the structural feature of IM30 dimer. I wonder of the relevance of this state in the context of the manuscript.*

Reply: We agree, in principle this manuscript contains two stories which could also have been submitted in two separate manuscripts. Yet, in our opinion only the two following two parts of the manuscript, carpet formation and the structural change upon ring disassembly, complete the story: IM30 rings, where the protomers are largely structured, disassemble on membrane surfaces, and this involves partly unfolding of the monomers. Clearly, currently we can not safely state which structure the monomers possess on the membrane surface, furthermore it is possible that the unstructured region restructures after ring dissociation and membrane binding. Analyzing and being able to describe aforementioned questions is a current task in the Schneider group.

Comment 9:

The above AFM (and FRET) data show that IM30 and IM30 disassemble on the membrane. In this case, the isolated oligomer is not the functional state. Any structure of the soluble oligomer cannot reveal how IM30* protect the membrane. A structure of IM30* bound to membrane would be more relevant (although much more challenging to obtain). I acknowledge how difficult this could be, but I would suggest solid-state NMR. But again, localizing further the regions of IM30 in interaction with the membrane (limited proteolysis) would be an original result.*

Reply: Actually, the AFM data show that wt IM30 rings disassemble on the membrane, whereas the dimeric IM30* rather assembles to form large carpets on the membrane surface. Furthermore, as the mutant with its disordered regions binds very well to membrane surfaces and is even more membrane-protective than the wt, one might hypothesize that the solution structure of the mutant resembles the one of the membrane-bound state. We actually believe that the inter-molecular interactions between the IDP regions, which we already see in the soluble IM30* dimer, are still present in the membrane carpet structure.

We completely agree that the structure of the soluble monomer cannot reveal how IM30^(*) protects the membrane. To show this, the structure of the membrane-bound protein is indeed required. Yet, as stated by this reviewer, this is far beyond the scope of the present manuscript. Nevertheless, it is a current project in the lab, and we hope to be able to describe the structure of the monomers on membrane surfaces in more detail in a couple of years. However, we would like to mention that it is not the aim of the present manuscript to present the structure of membrane-bound IM30 monomers. We merely show that (i) IM30 rings disassemble upon membrane binding plus that (ii) disassembly involves unfolding of about half of the protein and that (iii) the protein forms membrane-covering carpets upon ring dissociation. These points are well supported by the presented data in our opinion.

Comment 10:

The SAXS, CD, HDX and limited proteolysis on the full length IM30 all converge to conclude that helices 3b-7 is disordered. Nevertheless, this mutant has a quadruple mutation in helix 4, and these mutations are likely to destabilize an important folding core (in particular the F to A mutation). The fact that the mutant does not form large ring underpins the importance of this missing fold. Here, the disorder could well be an experimental artifact, and is not worth deep investigation such as extended modeling.*

Reply: We totally agree, introducing mutations into a protein, an approach well established in protein biochemistry for decades, always bears the risk of gaining or stabilizing a non-native fold. This, unfortunately, is a risk all protein researches have to accept when working with mutants.

However, we have replaced the FERM residues located in the predicted helix 4 by Ala residues, *i.e.*, by helix-stabilizing residues. Thus, we do not expect to reduce the helix-forming propensity of these sequence regions (rather, we probably increase it).

Helix 2 and helix 3 remain structured in the IM30* mutant, according to our CD+SAXS data, again disfavoring the idea that the mutations themselves induce disorder, whereas the isolated h3b-h7 fragment with wt-residues is disordered.

Figure: The residues F¹⁶⁹-M¹⁷¹ are located in helix 4 and might interact with the C-terminal part of helix 3 (helix 3b).

Importantly, the here analyzed truncated IM30 protein (IM30_H3b-7) contains a small part of helix 3 (helix 3b), and all (potential) interactions between the FERM residues and helix 3 involve residues in helix 3b, i.e., in the region which is included in the analyzed truncated protein (see structure model above). Thus, all interactions between the FERM residues and residues of other helices (of the same molecule) are possible, and a potential folding core is retained in the truncated protein. Yet, this construct does not show the expected α -helix content, arguing against the idea that the FERM residues are a “folding core”. We would like to note that we also analyzed a longer construct, where a larger part of helix 3 is retained, and also this construct (starting at aa 124, i.e. about half of the full helix 3) contains only the α -helix content we expected from the helix 3 contribution when analyzed via CD spectroscopy (not shown).

In summary, the mutations do not explain why the individual IM30 regions do not form proper α -helices anymore.

All these pieces of information indicate that different intermolecular interactions, involving FERM-mediated interactions, are crucial for oligomer formation. Thus, this reviewer very likely is correct in that we have identified and mutated residues, which are crucial for the formation of properly structured IM30 oligomers. Actually, this was exactly the reason why we mutated these residues. Clearly, without having a structure of IM30 monomers/rings, we can currently not definitely say whether the lack of ring formation is caused by abolished protomer-protomer contacts, which results in structuring of the monomer, or whether mutating the FERM residues reduced helix-helix contacts in the monomer, resulting in abolished supercomplex formation.

Finally, in a recently published “divide and conquer” approach (Ref. 30) analyzing various truncated IM30 WT fragments, we have already concluded that helix 4 is crucial for stabilizing higher-ordered IM30 oligomers. Thus, all our observations prompt us to conclude that we indeed mutated residues that are crucial for IM30 oligomerization and ring formation, i.e., a core crucial for assembly of the oligomer.

We have noticed that further explanation is urgently needed in the text, and we, therefore, discuss the potential folding of IM30 monomers/oligomers as well as the role of individual (predicted) helices now in detail in a new paragraph in the discussion on page 10 – 11, line 175-217.

Comment 11:

Line 108-109: The authors write: “The major structural difference between IM30 and IM30 WT lies in the region of the predicted helices 1, 3b, 4, and 5/6”, which for me indicates that these regions are NOT involved in membrane binding, but more likely in ring formation. On the contrary, the fact that the mutant keeps a high affinity for membranes suggest that the activity is localized at the H2-3a coiled-coil. Coiled-coil regions are known to have affinity for membrane.*

Reply: The reviewer is absolutely correct, and we apologize for obviously not having stated this unmistakably. The H2-3 coiled-coil is indeed mainly involved in membrane interactions, as we have previously shown (Ref. 30). Our data also indicate that the C-terminal region is critical for formation of higher oligomers, which involves ring formation in solution and carpet formation on membrane surfaces.

As this is an important point, we now elaborate in more detail on said subject, hence we have added another paragraph in the discussion, addressing this issue (page 10 – 11, line 175-217)

Comment 12:

The NMR data on the truncated construct cannot be conclusive unless the author validate the ‘divide and conquer’ approach by showing that the NMR spectrum of the full-length oligomer super-imposed with that of H2-3b + H1 + H3b-7. Truncation of H3b-7 can lead to unfolding of this truncated product.

Reply: Actually, we already performed (and have published, Ref. 30) a rather extensive “divide and conquer” approach using various IM30 proteins where individual helices are deleted. In the said study, we recognized for the first time that the simple addition of the spectra 1-3 + 4-7 does not result in the spectrum of the WT protein, which is incorporated in rings. It actually is exactly one major point of the current manuscript that the full-length protein is highly α -helical when organized in rings, whereas about half of it is unstructured when the ring disassembles. Furthermore, we would like to mention that the region containing helices 4-7 is also unstructured in the context of the not truncated, full-length IM30* protein. Finally, in the past we have indeed tried to obtain structural information via NMR using the full-length WT protein but failed as it was impossible to separate intra-molecular interactions from inter-molecular interactions in the homo-oligomeric rings.

Comment 13:

From the modelling refined with the SAXS data, the authors conclude that the C-terminal tail is disordered but is the dimeric interface. There is an apparent contradiction here, and to-date, only few example of peptides are involved in an interaction interface while remaining disordered (absence of folding upon binding, for example the study by Borgia et al, Nature, 2018, 555 p 61-68 on the histone H1 tail). To be convincing, other experimental data should be added, such as NMR (chemical shift and PRE-NMR), FRET, fluorescent anisotropy, etc...

Reply: In the present manuscript, we show dimerization of the intrinsically disordered C-terminal regions of IM30 using three different approaches: (1) SAXS, (2) size exclusion chromatography, and (3) oxidative Cys cross-linking. Potentially, this reviewer has overlooked the SEC data shown in the supplementary Figure (Supplementary Fig. 13). We completely agree that a statement solely based on the SAXS data might be weak. Yet, in our opinion, when three different methods are applied which all show that interactions are mediated via the C-terminus, it is fair to make this statement. Thus, as requested by this reviewer, we have shown that oligomerization is mediated by the C-terminus via using different techniques.

Comment 14:

Also, the addition of cysteine before SDS-PAGE cannot be conclusive for the presence of dimer in the native state. Upon SDS addition and denaturation, any cysteine is available for disulfide-bound formation, whether it was in a folded domain or not in the native state. Native mass spectrometry would be more conclusive.

Reply: Oxidative Cys cross-linking is a technique well established in protein biochemistry for decades. Actually, it has turned out (from the beginning) that disulfides do form only when residues are in close proximity in a structure, and interactions are typically mediated by surrounding residues. When an (unfolded) protein is analyzed, e.g. in SDS sample buffer, disulfide bridges do not form. We have performed native MS in the past (Ref. 19), and based on our experience and on what we see in the literature, native MS is far more problematic and prone to artifacts than other techniques.

Comment 15:

Finally, the authors mention liquid-liquid phase separation and membrane-less organelles in their conclusion. This seems out of the topic of this study, as IM30 is a membrane-associating protein.

Reply: We agree, this passage probably was a bit out of context. Therefore, we rephrased it, thus it is more intelligible for the reader. We now write on page 12-13, line 238-244:

“In fact, dynamic self-assembly is typically observed with IDPs, often involving liquid-liquid phase separation^{33,41,42}. In contrast to the formation of membrane-less organelles in cells, induced by liquid-liquid phase separation of IDPs, demixing into a condensed and a protein-light fraction (*i.e.* carpets and unassociated but membrane-attached protomers) appears to take place on the membrane surface in case of IM30. Restricting protein-protein interaction to the membrane surface limits the degrees of freedom to a 2D surface, which likely increased the efficiency of carpet formation.”

REVIEWERS' COMMENTS:

Reviewer #1 (Remarks to the Author):

The authors could clarify all relevant critical aspects in their detailed responses to the reviewers and they have improved their manuscript accordingly.

I therefore recommend publication of this manuscript.

Reviewer #2 (Remarks to the Author):

The authors have greatly improved the manuscript, and which is now easier to read. I repeat that it contains a wealth of data, and these are now very clearly presented. The claims of the author:

- 1) IM30 forms a protecting carpet on lipid membranes,
 - 2) H2/3 is involved in membrane-binding and
 - 3) H3b-H7 is disordered and is involved in the protein-protein interactions
- are convincingly presented.

In particular, the coherence of the manuscript and the connection between the two main parts of the manuscript is now fluid and straightforward.

About the particular comments I had in the previous version:

A- The authors convinced me that their AFM data are robust (and that HEPES is the right choice of buffer). The controls are convincing (IM30wt and IM30* on mica without lipids in HEPES). The latter is included in their answer to the referee, and it could be in the Supplementary material figure 4 (together with the control IM30wt) or in Supp Mat figure 5.

B- Also, I think their clarification about the choice of buffer in their answer to my comment 4 could also be included in page 6 of the Supp Material. This would clarify the Supp Figures 3 and 4, which - at first- looked a bit out of place to me.

“In previous experiments, we have found that the protein is not able to fulfill most of the described in-vitro activities in the presence of TRIS. In particular, IM30 does not bind to membrane surfaces anymore. Whilst this buffer is not relevant for the study of the conformation of IM30 on membrane, it can be used to visualize IM30 rings... (Sup Figure 3).”

C- I had indeed overlooked their very nice SEC-MALS data. These are convincing. I also had overlook their previous divide and conquer approach that is already published.

D- My last remark would concern the cross-linking. I agree that it is a well-established technique, and that it is appropriate to be used here. Nevertheless, I maintain that SDS-PAGE is not enough to probe for the disulphide-bound (or absence of) on the dimer interface, because of the SDS denaturation before electrophoresis.

In order to show the disulphide-bound under native conditions, I actually should have mentioned MALDI (and not native mass-spec), or native-PAGE. I apologies for this mistake in my first comments. The absence of free cysteines can also be shown using colorimetric agents such as the Ellman's reagent (DTNB) – several methods are presented in: Winther JR, Thorpe C. Quantification of thiols and disulfides. *Biochim Biophys Acta*. 2014;1840(2):838-846. doi:10.1016/j.bbagen.2013.03.031

Finally, I deeply acknowledge the efforts of the author, the manuscript present novel and significant discovery – membrane protection related to plant stress – mediated by a highly dynamic protein with an original mechanism of action that might be generalized to other biological process. This is very significant and should be published.

Response to reviewers

We appreciate the reviewers' constructive remarks that have enabled us to further improve our manuscript "*IM30 IDPs form a membrane protective carpet upon supercomplex disassembly*". Below, the final remarks of reviewer #2 are reproduced in *italics*; our reply and the resulting changes to the manuscript appear in regular font.

Reviewer #2:

The authors have greatly improved the manuscript, and which is now easier to read. I repeat that it contains a wealth of data, and these are now very clearly presented. The claims of the author:

- 1) IM30 forms a protecting carpet on lipid membranes,*
- 2) H2/3 is involved in membrane-binding and*
- 3) H3b-H7 is disordered and is involved in the protein-protein interactions are convincingly presented.*

In particular, the coherence of the manuscript and the connection between the two main parts of the manuscript is now fluid and straightforward.

Reply: We thank this reviewer for this very positive comment.

About the particular comments I had in the previous version:

A- The authors convinced me that their AFM data are robust (and that HEPES is the right chose of buffer). The controls are convincing (IM30wt and IM30 on mica without lipids in HEPES). The latter is included in their answer to the referee, and it could be in the Supplementary material figure 4 (together with the control IM30wt) or in Supp Mat figure 5.*

Reply: As suggested, we now added the figure from the (previous) response letter as new Fig. S4d. To keep the dimensions of the entire figure, the height profile shown before as Fig. S4d is now given as an inset in Fig. S4c.

B- Also, I think their clarification about the choice of buffer in their answer to my comment 4 could also be included in page 6 of the Supp Material. This would clarify the Supp Figures 3 and 4, which -at first- looked a bit out of place to me.

"In previous experiments, we have found that the protein is not able to fulfill most of the described in-vitro activities in the presence of TRIS. In particular, IM30 does not bind to membrane surfaces anymore. Whilst this buffer is not relevant for the study of the conformation of IM30 on membrane, it can be used to visualize IM30 rings... (Sup Figure 3)."

Reply: As suggested, we now write on page 7 of the Supplement: “Furthermore, we have found in previous experiments that the protein is not able to fulfill most of the described *in vitro* activities in the presence of Tris. In particular, IM30 does not bind to membrane surfaces anymore. Yet, while Tris buffer is not relevant for studying the conformation of IM30 on membrane surfaces, our results show that AFM is capable of imaging IM30 with resolutions that allow discrimination between the prototypical ring structures and deviations from that (Sup Fig. 3 and 4).”

C- I had indeed overlooked their very nice SEC-MALS data. These are convincing. I also had overlook their previous divide and conquer approach that is already published.

Reply: We appreciate this positive comment.

*D- My last remark would concern the cross-linking. I agree that it is a well-established technique, and that it is appropriate to be used here. Nevertheless, I maintain that SDS-PAGE is not enough to probe for the disulphide-bound (or absence of) on the dimer interface, because of the SDS denaturation before electrophoresis. In order to show the disulphide-bound under native conditions, I actually should have mentioned MALDI (and not native mass-spec), or native-PAGE. I apologies for this mistake in my first comments. The absence of free cysteines can also be shown using colorimetric agents such as the Ellman's reagent (DTNB) – several methods are presented in: Winther JR, Thorpe C. Quantification of thiols and disulfides. *Biochim Biophys Acta*. 2014;1840(2):838-846. doi:10.1016/j.bbagen.2013.03.031*

Reply: We apologize, obviously we have not stated perfectly clear that we did not intend to probe disulphide-bound formation under native conditions, i.e. within cells. The disulphide brides will obviously form only after the cells are broken. This method is merely used to show neighborhood of the cross-linked cysteines, a further indication for interaction (as shown by various methods in the manuscript). The protein being denatured prior to SDS-PAGE, and then consequently not able to form disulphide brides anymore, is exactly the advantage of the present method (rather than a problem). To clarify this, we now write on page 9: “This mutant ran as a dimer on SDS gels after purification (Fig. 3c), ...”. “after purification” was added.

Finally, I deeply acknowledge the efforts of the author, the manuscript present novel and significant discovery – membrane protection related to plant stress – mediated by a highly dynamic protein with an original mechanism of action that might be generalized to other biological process. This is very significant and should be published.

Reply: We highly appreciate this positive comment and thank the reviewer for his/her constructive remarks that have definitely assisted in significantly improving the article.